# Neural Clamping: Joint Input Perturbation and Temperature Scaling for Neural Network Calibration

**Yung-Chen Tang**                                    *yctang@cse.cuhk.edu.hk*
*The Chinese University of Hong Kong*
*National Tsing Hua University*

**Pin-Yu Chen**                                       *pin-yu.chen@ibm.com*
*IBM Research*

**Tsung-Yi Ho**                                       *tyho@cse.cuhk.edu.hk*
*The Chinese University of Hong Kong*

**Reviewed on OpenReview:** *https://openreview.net/forum?id=qSFToMqLcq*

## Abstract

Neural network calibration is an essential task in deep learning to ensure consistency between the confidence of model prediction and the true correctness likelihood. In this paper, we propose a new post-processing calibration method called **Neural Clamping**, which employs a simple joint input-output transformation on a pre-trained classifier via a learnable universal input perturbation and an output temperature scaling parameter. Moreover, we provide theoretical explanations on why Neural Clamping is provably better than temperature scaling. Evaluated on BloodMNIST, CIFAR-100, and ImageNet image recognition datasets and a variety of deep neural network models, our empirical results show that Neural Clamping significantly outperforms state-of-the-art post-processing calibration methods. The code is available at github.com/yungchentang/NCToolkit, and the demo is available at huggingface.co/spaces/TrustSafeAI/NCTV.

## 1 Introduction

Deep neural networks have been widely deployed in real-world machine learning empowered applications such as computer vision, natural language processing, and robotics. However, without further calibration, model prediction confidence usually deviates from the true correctness likelihood (Guo et al., 2017). The issue of poor calibration in neural networks is further amplified in high-stakes or safety-critical decision making scenarios requiring accurate uncertainty quantification and estimation, such as disease diagnosis (Jiang et al., 2012; Esteva et al., 2017) and traffic sign recognition systems in autonomous vehicles (Shafaei et al., 2018). Therefore, calibration plays an important role in trustworthy machine learning (Guo et al., 2017; Kumar et al., 2019; Minderer et al., 2021).

Recent studies on neural network calibration can be mainly divided into two categories: *in-processing* and *post-processing*. In-processing involves training or fine-tuning neural networks to mitigate their calibration errors, such as in Müller et al. (2019); Liang et al. (2020); Tian et al. (2021); Qin et al. (2021); Tao et al. (2023). Post-processing involves post-hoc intervention on a pre-trained neural network model without changing the given model parameters, such as adjusting the data representations of the penultimate layer (i.e., the logits) to calibrate the final softmax layer's output of prediction probability estimates. As in-processing calibration tends to be time-consuming and computationally expensive, in this paper, we opt to focus on post-processing calibration.

Current post-processing calibration methods are predominately shed on processing or remapping the output logits of neural networks, *e.g.,* Guo et al. (2017); Kull et al. (2019); Gupta et al. (2020); Tian et al. (2021);

Xiong et al. (2023). However, we aim to provide a new perspective and show that joint input-output model calibration can further improve neural network calibration. The rationale is that active adjustment of data inputs will affect their representations in every subsequent layer, rather than passive modification of the output logits.

In this paper, we propose a new post-processing calibration framework for neural networks. We name this framework **Neural Clamping** because its methodology is based on learning a simple joint input-output transformation for calibrating a pre-trained (frozen) neural network classifier. Figure 1 illustrates the entire procedure of Neural Clamping. We consider a $K$-way neural network classifier $f_\theta(\cdot) \in \mathbb{R}^K$ with fixed model parameters $\theta$. The classifier outputs the logits for $K$ classes and uses softmax on the logits to obtain the final confidence on class predictions (i.e., probability scores). To realize joint input-output calibration, Neural Clamping adds a trainable universal perturbation $\delta$ to every data input and a trainable temperature scaling parameter $T$ at the output logits. The parameters $\delta$ and $T$ are jointly learned by minimizing the focal loss Lin et al. (2017) with a weight-decay regularization term trained on a calibration set (i.e. validation set) $\{x_i, y_i\}_{i=1}^n$ for calibration. The focal loss assigns non-uniform importance on $\{x_i\}_{i=1}^n$ during training and includes the standard cross entropy loss as a special case. Finally, in the evaluation (testing) phase, Neural Clamping appends the optimized calibration parameters $\delta^*$ and $T^*$ to the input and output of the fixed classifier $f_\theta(\cdot)$, respectively.

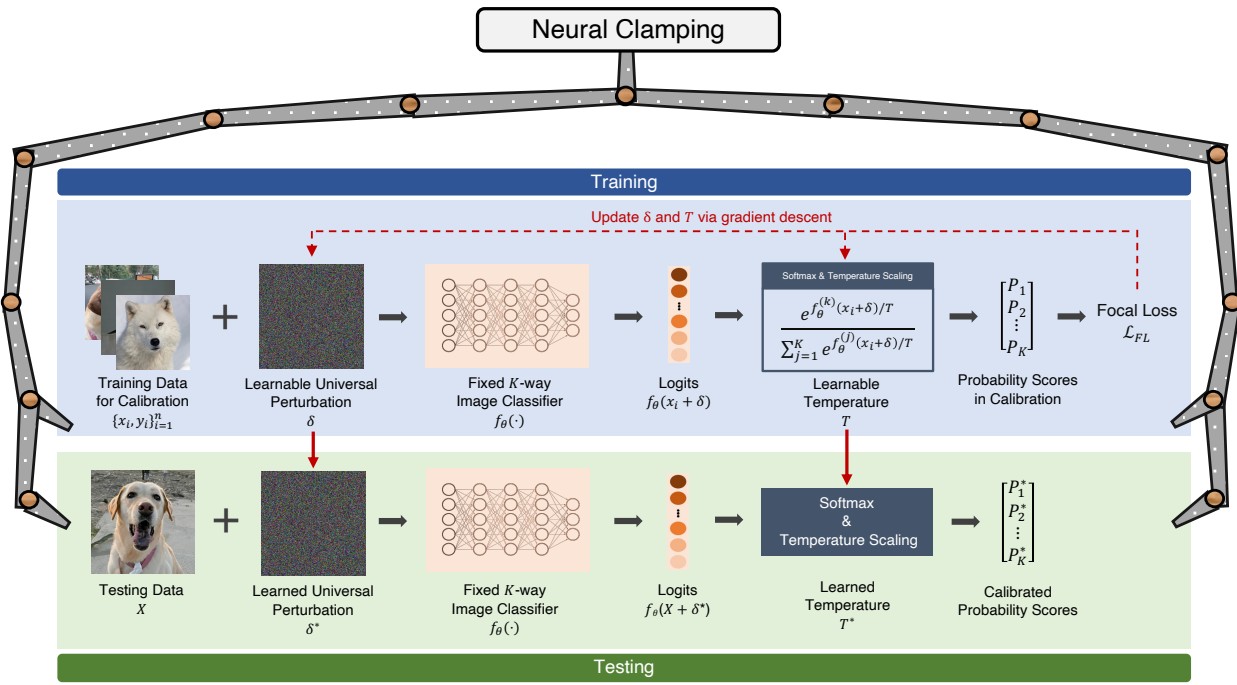

Figure 1: Overview of Neural Clamping: a joint input-output post-processing calibration framework.

Our main contributions are summarized as follows:

- We propose Neural Clamping as a novel joint input-output post-processing calibration framework for neural networks. Neural Clamping learns a universal input perturbation and a temperature scaling parameter at the model output for calibration. It includes temperature scaling as a special case, which is a strong baseline for post-processing calibration.

- We develop theoretical results to prove that Neural Clamping is better than temperature scaling in terms of constrained entropy maximization for uncertainty quantification. In addition, we use first-order approximation to optimize the data-driven initialization term for the input perturbation, improving the stability of Neural Clamping in our ablation study. Furthermore, we leverage this theoretical result to design a computationally efficient algorithm for Neural Clamping.

- Evaluated on different deep neural network classifiers (including ResNet (He et al., 2016), Vision Transformers (Dosovitskiy et al., 2020), and MLP-Mixer (Tolstikhin et al., 2021)) trained on Blood-MNIST (Yang et al., 2023), CIFAR-100 (Krizhevsky et al., 2009), and ImageNet-1K (Deng et al., 2009) datasets and three calibration metrics, Neural Clamping outperforms state-of-the-art post-processing calibration methods. For instance, when calibrating the ResNet-110 model on CIFAR-100, the expected calibration error is improved by 34% when compared to the best baseline.

## 2 Background and Related Work

In this section, we begin by introducing the problem formulation for calibration and describing the notations used in this paper. Furthermore, we define different metrics used to measure calibration error and conclude this section with an overview of the post-processing calibration methods.

### 2.1 Probabilistic Characterization of Neural Network Calibration

Assume a pair of data sample and label $(\boldsymbol{x}, y)$ is drawn from a joint distribution $\mathcal{D} \subset \mathbb{X} \times \mathbb{Y}$, where $\boldsymbol{x} \in \mathbb{X}$ is a data sample, and $y \in \mathbb{Y} = \{1, ..., K\}$ is the ground-truth class label. Let $f_\theta : \mathcal{X} \to \mathbb{R}^K$ denote a $K$-way neural network classifier parametrized by $\theta$, where $\boldsymbol{z} = f_\theta(\boldsymbol{x}) = [z_1, \ldots, z_K]$ is the model's output *logits* of a given data input $\boldsymbol{x}$. Following the convention of neural network implementations, the prediction probability score of $\boldsymbol{x}$ is obtained by applying the *softmax* function $\sigma$ on $\boldsymbol{z}$, denoted as $\sigma(\boldsymbol{z}) \in [0,1]^K$. The $k$-th component of $\sigma(\boldsymbol{z})$ is defined as $\sigma(\boldsymbol{z})_k = \exp(\boldsymbol{z}_k)/\sum_{k'=1}^K \exp(\boldsymbol{z}_{k'})$, which satisfies $\sum_{k=1}^K \sigma(\boldsymbol{z})_k = 1$ and $\sigma(\boldsymbol{z})_k \geq 0$. Suppose the model predicts a most likely class $\hat{y} = \arg\max_k \sigma(\boldsymbol{z})_k$ with confidence $\hat{p} = \sigma(\boldsymbol{z})_{\hat{\boldsymbol{y}}}$. Formally, the model $f_\theta$ is called *calibrated* if

$$\mathbb{P}(\mathrm{y} = \hat{y}|\mathrm{p} = \hat{p}) = \hat{p}, \tag{1}$$

where $\mathbb{P}$ denotes probability and p denotes the true likelihood. Equation (1) only considers the prediction confidence of the most likely (top-1) class. We can extend it to consider the prediction confidence of every class. Let the class-wise prediction confidence be $\hat{p}_i = \sigma(\boldsymbol{z})_i$ for $i = \{1, ..., K\}$, the network is called *classwise-calibrated* if

$$\mathbb{P}(\mathrm{y} = i|\mathrm{p}_i = \hat{p}_i) = \hat{p}_i, \quad \forall \, i \in \mathbb{Y} \tag{2}$$

where $p_i$ is the true likelihood for class $i$.

### 2.2 Calibration Metrics

**Expected Calibration Error (ECE).** Calibration error aims to compute the difference between confidence and accuracy as

$$\mathbb{E}_{(x,y)\sim\mathcal{D}} \left[ |\mathbb{P}(\mathrm{y} = \hat{y}|\mathrm{p} = \hat{p}) - \hat{p}| \right] \tag{3}$$

Unfortunately, this quantity cannot be exactly computed from equation (3) if the underlying data distribution $\mathbb{D}$ is unknown. The most popular metric to measure calibration is the *Expected Calibration Error* (ECE) (Guo et al., 2017; Naeini et al., 2015). ECE approximates the calibration error by partitioning predictions into $m$ intervals (bins) $\{B_i\}_{i=1}^m$. The calibration error is calculated by first taking the difference between the confidence and accuracy in each bin and then computing the weighted average across all bins, i.e.,

$$\mathrm{ECE} = \sum_{i=1}^M \frac{|B_i|}{n} |\mathsf{acc}(B_i) - \mathsf{conf}(B_i)| \tag{4}$$

where $|B_i|$ is the number of samples in bin $B_i$, $n$ is the total number of data, and $\mathsf{acc}(B_i)$ and $\mathsf{conf}(B_i)$ is the accuracy and confidence in $B_i$, respectively.

**Adaptive Expected Calibration Error (AECE) (Mukhoti et al., 2020).** Since most data for a trained model fall into the highest confidence bins, these bins mostly determine the value of the ECE. Instead of pre-defined intervals for bin partitioning, in AECE, adaptive interval ranges ensure each bin has the same number of samples. AECE is defined as

$$\text{AECE} = \sum_{i=1}^{M} \frac{|B_i|}{n} |\mathsf{acc}(B_i) - \mathsf{conf}(B_i)| \tag{5}$$
$$\text{subject to} \quad |B_i| = |B_j| \quad \forall i, j$$

where $|B_i|$ is the number of samples in bin $B_i$, $n$ is the total number of data, and $\mathsf{acc}(B_i)$ and $\mathsf{conf}(B_i)$ is the accuracy and confidence in $B_i$, respectively.

**Static Calibration Error (SCE) (Nixon et al., 2019).** ECE does not take into account the calibration error for all classes. It only calculates the calibration error of the top-1 class prediction. SCE extends ECE and considers multi-class predictions based on Equation (2):

$$\text{SCE} = \frac{1}{K} \sum_{k=1}^{K} \sum_{i=1}^{M} \frac{|B_i^k|}{n} |\mathsf{acc}(i,k) - \mathsf{conf}(i,k)| \tag{6}$$

where $K$ is the number of classes, $|B_i^k|$ is the number of samples in bin $i$ of class $k$, $n$ is the total number of data, and $\mathsf{acc}(i,k)$ and $\mathsf{conf}(i,k)$ is the accuracy and confidence in $B_i^k$, respectively.

## 2.3 Post-Processing Calibration Methods

**Temperature Scaling (Guo et al., 2017).** Temperature scaling is the simplest variant of Platt scaling (Platt et al., 1999), which is a method of converting a classifier's output into a probability distribution over all classes. Specifically, all classes have the same scalar parameter (i.e., temperature) $T > 0$ in the softmax output such that $\hat{\boldsymbol{q}} = \sigma(\boldsymbol{z}/T)$, where $\hat{\boldsymbol{q}} \in [0,1]^K$ denotes the calibrated probability scores. It is worth noting that by definition temperature scaling only changes the confidence but not the class prediction. Moreover, the entropy of $\hat{\boldsymbol{q}}$ increases with $T$ when $T \geq 1$. The temperature $T$ is optimized via the Negative Log Likelihood (NLL) over a calibration training set $\{\boldsymbol{x_i}\}_{i=1}^{n}$, where NLL is defined as $-\sum_{i=1}^{n} \log(\hat{q}_{i,y_i})$ and $\hat{q}_{i,y_i}$ is the prediction on the correct class $y_i$ for the $i$-th sample.

**Vector Scaling and Matrix Scaling (Guo et al., 2017).** They are two extensions of Plat scaling (Platt et al., 1999). Let $\boldsymbol{z}$ be the output logits for an input $\boldsymbol{x}$. Vector Scaling and Matrix Scaling adopt linear transformations on $\boldsymbol{z}$ such that $\hat{\boldsymbol{q}} = \sigma(\boldsymbol{W}\boldsymbol{z} + \boldsymbol{b})$, where $\boldsymbol{W} \in \mathbb{R}^{K \times K}$ and $\boldsymbol{b} \in \mathbb{R}^K$ for both settings. Vector scaling is a variation of matrix scaling when $\boldsymbol{W}$ is restricted to be a diagonal matrix. The parameters $\boldsymbol{W}$ and $\boldsymbol{b}$ are optimized based on NLL.

**MS-ODIR and Dir-ODIR (Kull et al., 2019).** The authors in Kull et al. (2019) proposed Dirichlet calibration and ODIR term (Off-Diagonal and Intercept Regularization). The difference between matrix scaling and Dirichlet calibration is that the former affects logits while the latter modifies pseudo-logits through $\hat{\boldsymbol{q}} = \sigma(\boldsymbol{W} \ln(\sigma(\boldsymbol{z})) + \boldsymbol{b})$, where $\ln(\cdot)$ is a component-wise natural logarithm function. The results in Guo et al. (2017) indicated poor matrix scaling performance, because calibration methods with a large number of parameters will over-fit to a small calibration set. Therefore, Kull et al. (2019) proposed a new regularization method called ODIR to address the overfitting problem, i.e. $\mathsf{ODIR} = \frac{1}{K(K-1)} \sum_{j \neq j} w_{j,k} + \frac{1}{K} \sum_j b_j$, where $w_{j,k}$ and $b_j$ are elements of $\boldsymbol{W}$ and $\boldsymbol{b}$, respectively. MS-ODIR applies Matrix Scaling (Guo et al., 2017) with ODIR, while Dir-ODIR uses Dirichlet calibration (Kull et al., 2019).

**Spline-Fitting (Gupta et al., 2020) and Density-Ratio Calibration (Xiong et al., 2023).** The Spline-fitting approximates the empirical cumulative distribution with splines to re-calibrate network outputs for calibrated probabilities. However, this method is limited to calibration on the top-$k$ class prediction but cannot be extended to all-class predictions. The Density-Ratio Calibration approach focuses on estimating continuous density functions, which aligns well with calibration methods that produce continuous confidence scores, such as scaling-based calibration techniques. However, this method is limited in that it can only calibrate the predicted probabilities, and cannot calibrate the full probability distribution output by the model. Therefore, these methods are beyond our studied problem of all-class post-processing calibration.

## 3 Neural Clamping

Based on the proposed framework of Neural Clamping as illustrated in Figure 1, in this section we provide detailed descriptions on the joint input-output calibration methodology, the training objective function, the influence of hyperparameter selection, and the theoretical justification on the improvement over temperature scaling and the date-driven initialization.

### 3.1 Joint Input-Output Calibration

To realize joint input-output calibration, Neural Clamping appends a learnable universal perturbation $\delta$ at the model input and a learnable temperature scaling parameter $T$ for all classes at the model output. In contrast to the convention of output calibration, Neural Clamping introduces the notion of input calibration by applying simple trainable transformations on the data inputs prior to feeding them to the model. In our implementation, the input calibration is simply a universal additive perturbation $\delta$. Therefore, Neural Clamping includes temperature scaling as a special case when setting $\delta = 0$.

Modern neural networks often suffer from an overconfident issue, resulting in a lack of well-calibration and low output entropy (Guo et al., 2017; Mukhoti et al., 2020). Calibrating neural networks, a common approach to address the overconfident issue, typically results in increased entropy as a byproduct. In the seminal work on neural network calibration by Guo et al. (2017), it is noted that adjusting the temperature parameter to improve calibration aligns with the objective of maximizing the entropy of the output probability distribution under additional constraints.

Building upon this notion, we extend the problem formulation presented in Guo et al. (2017), which utilizes entropy to study output calibration. We evaluate this problem on a calibration set $\{\boldsymbol{x}_i, y_i\}_{i=1}^n$ with an input perturbation $\boldsymbol{\delta}$. The objective is to find the best-calibrated output $q^*$ that maximizes the entropy of $q^*$ while satisfying the specified calibration constraints:

$$\text{Maximize}_{q \in \mathbb{R}^K} - \sum_{i=1}^n q(\boldsymbol{z}_i)^\top \log(q(\boldsymbol{z}_i))$$
$$\text{subject to } q(\boldsymbol{z}_i)^{(k)} \geq 0 \quad \forall i \in \{1, \ldots, n\} \text{ and } k \in \{1, \ldots, K\} \tag{7}$$
$$\sum_{i=1}^n \mathbf{1}^\top q(\boldsymbol{z}_i) = 1$$
$$\sum_{i=1}^n \boldsymbol{z}_i^\top \boldsymbol{e}^{(y_i)} = \sum_{i=1}^n z_i^\top q(\boldsymbol{z}_i)$$

where $\cdot^\top$ denotes vector transpose, $\boldsymbol{z}_i = f_\theta(\boldsymbol{x}_i + \delta)$ is the logit of $\boldsymbol{x}_i + \boldsymbol{\delta}$, $\mathbf{1}$ is an all-one vector, $\boldsymbol{e}^{(y_i)}$ is an one-hot vector corresponding to the class label $y_i$, and $\log(\cdot)$ is an element-wise log operator. The first two constraints guarantee that $q$ is a probability distribution, whereas the third constraint restricts the range of possible distributions, which stipulates that the average true class logit equals to the average weighted logit.

To motivate the utility of joint input-output calibration, the following lemma formally states that the proposed form of joint input perturbation and temperature scaling in Neural Clamping is the unique solution $q^*$ to the above constrained entropy maximization problem.

**Lemma 3.1** (optimality of joint input-output calibration). *For any input perturbation $\boldsymbol{\delta}$, let $f_\theta(\cdot) = [f_\theta^{(1)}, \ldots, f_\theta^{(K)}]$ be a fixed $K$-way neural network classifier and let $\boldsymbol{z}$ be the output logits of a perturbed data input $\boldsymbol{x} + \boldsymbol{\delta}$. Then the proposed form of joint input-output calibration in Neural Clamping is the unique solution $q^*(z)^{(k)} = \frac{\exp[f_\theta^{(k)}(\boldsymbol{x}+\boldsymbol{\delta})/T]}{\sum_{j=1}^K \exp[f_\theta^{(j)}(\boldsymbol{x}+\boldsymbol{\delta})/T]}, \forall k \in \{1, \ldots, K\}$, to the constrained entropy maximization problem in equation 7.*

*Proof.* The proof is given in Appendix B.

### 3.2 Training Objective Function in Neural Clamping

Neural Clamping uses the focal loss (Lin et al., 2017) and a weight-decay regularization term as the overall objective function for calibration. It has been shown that the focal loss is an upper bound of the regularized

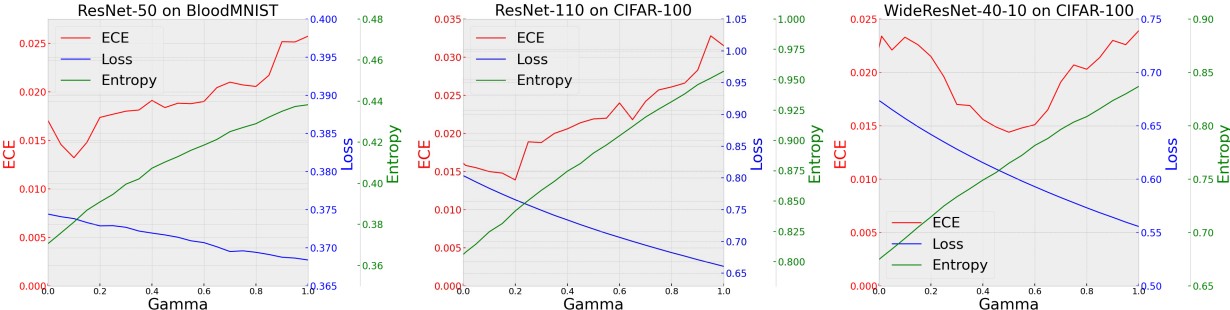

Figure 2: Neural Clamping on ResNet-50/ResNet-110 and Wide-ResNet-40-10 with different $\gamma$ values and the resulting expected calibration error (ECE), training loss, and entropy on BloodMNIST and CIFAR-100. When $\gamma = 0$, focal loss reduces to cross entropy loss. The experiment setup is the same as Section 4.

KL-divergence (Charoenphakdee et al., 2021; Mukhoti et al., 2020). Therefore, minimizing the focal loss aims to reduce the KL divergence between the groundtruth distribution and the predicted distribution while increasing the entropy of the predicted distribution. Focal loss is an adjusted cross entropy loss with a modulating factor $(1 - \hat{p}_{i,y_i})^\gamma$ and $\gamma \geq 0$, where $\hat{p}_{i,y_i}$ is the prediction probability given by a neural network on the correct class $y_i$ for the $i$-th sample. When $\gamma = 0$, focal loss reduces to the standard cross entropy loss. Formally, it is defined as

$$\mathcal{L}_{FL}^\gamma(f_\theta(\boldsymbol{x}_i), y_i) = -(1 - \hat{p}_{i,y_i})^\gamma \log(\hat{p}_{i,y_i}) \tag{8}$$

Given a calibration training set $\{\boldsymbol{x}_i, y_i\}_{i=1}^n$, the optimal calibration parameters $\boldsymbol{\delta}$ and $T$ in Neural Clamping are obtained by solving

$$\boldsymbol{\delta^*}, T^* = \arg\min_{\boldsymbol{\delta}, T} \sum_{i=1}^n \mathcal{L}_{FL}^\gamma(f_\theta(\boldsymbol{x_i} + \boldsymbol{\delta})/T, y_i) + \lambda \|\boldsymbol{\delta}\|_2^2 \tag{9}$$

Like other post-processing calibration methods, Neural Clamping only appends a perturbation at the model input and a temperature scaling parameter at the model output. It does not require any alternations on the given neural network for calibration.

### 3.3 How to Choose a Proper $\gamma$ Value in Focal Loss for Neural Clamping?

The focal loss has a hyperparameter $\gamma$ governing the assigned importance on each data sample in the aggregated loss. To understand its influence on calibration, in Figure 2 we performed a grid search of $\gamma$ value between 0 and 1 with an interval of 0.05 to calibrate Wide-Resnet-40-10 (Zagoruyko & Komodakis, 2016) and DenseNet-121 (Huang et al., 2017) models trained on CIFAR-100 (Krizhevsky et al., 2009) dataset. While the entropy continues to increase as the gamma value increases, ECE attains its minimum at some intermediate $\gamma$ value and is better than the ECE of using cross entropy loss (i.e., $\gamma = 0$). This observation verifies the importance of using focal loss for calibration. In our implementation, we select the best $\gamma$ value that minimizes ECE of the calibration dataset from a candidate pool of $\gamma$ values with separate runs.

### 3.4 Theoretical Justification on the Advantage of Neural Clamping

Here we use the entropy after calibration as a quantifiable metric to prove that Neural Clamping can further increase this quantity over temperature scaling. Note that temperature scaling is a special case of Neural Clamping when there is no input calibration (i.e., setting $\delta = 0$). For ease of understanding, we define $g_i$ as the gradient of the output entropy $H(\sigma(f_\theta(\cdot)/T))$ with respect to the input data $\boldsymbol{x}_i = [x_i^{(1)}, \ldots, x_i^{(m)}] \in [\alpha, \beta]$, where $[\alpha, \beta] \subset \mathbb{R}^m \times \mathbb{R}^m$ means the bounded range of all feasible data inputs (e.g., every image pixel value is within $[0, 255]$). We further define $\ell \in \mathbb{R}^m$ and $\mu \in \mathbb{R}^m$ as the lower bound and the upper bound over all calibration data $\{x_i\}_{i=1}^n$ on each input dimension. That is, their $j$-th entry is defined as $\ell_j = \min_{i \in \{1,\ldots,n\}} x_i^{(j)}$ and $\mu_j = \max_{i \in \{1,\ldots,n\}} x_i^{(j)}$, respectively.

With the use of first-order approximation, the following theorem shows that given the same temperature value $T$, Neural Clamping increases the entropy of temperature scaling by $\boldsymbol{\delta}^\top \boldsymbol{g}$, demonstrating the advantage of involving input calibration. Furthermore, based on our derivation and the data-driven bounds $\ell$ and $\mu$, we can obtain a closed-form first-order optimal solution $\widetilde{\boldsymbol{\delta}}$ for maximizing the entropy increment $\boldsymbol{\delta}^\top \boldsymbol{g}$. We call $\widetilde{\boldsymbol{\delta}}$ the *data-driven initialization* for the input perturbation $\boldsymbol{\delta}$. We will perform an ablation study to compare the performance and stability of data-driven versus random initialization in Section 4.3. In the following theorem, the notation $|\cdot|$, sign, and $\odot$ denote element-wise absolute value, sign operation (i.e., $\pm 1$), and product (i.e., Hadamard product), respectively.

**Theorem 3.2. (provable entropy increment and data-driven initialization)** *Let $[\boldsymbol{\alpha}, \boldsymbol{\beta}]$ be the feasible range of data inputs and $g = \sum_{i=1}^{n} g_i = [g^{(1)}, \ldots, g^{(K)}]$ be the sum of local input gradients. Define $\boldsymbol{\eta} \in \mathbb{R}^m$ element-wise such that $\eta_j = \ell_j - \alpha_j$ if $g^{(j)} < 0$, $\eta_j = \beta_j - \mu_j$ if $g^{(j)} > 0$, and $\eta_j = 0$ otherwise, for every $j \in \{1, \ldots, m\}$. Approaching by first-order approximation and given the same temperature value $T$, Neural Clamping increases the entropy of temperature scaling by $\boldsymbol{\delta}^\top \boldsymbol{g}$. Furthermore, the optimal value $\widetilde{\boldsymbol{\delta}}$ for maximizing $\boldsymbol{\delta}^\top \boldsymbol{g}$ is $\widetilde{\boldsymbol{\delta}} = \text{sign}(\boldsymbol{g}) \odot \boldsymbol{\eta}$.*

*Proof.* The proof is given in Appendix C.

Table 1: Comparison with various calibration methods on BloodMNIST with ResNet-50. The reported results are mean and standard deviation over 5 runs. The best/second-best method is highlighted by blue/green color. On ECE/AECE, the relative improvement of Neural Clamping to the best baseline is 31% and 28%, respectively.

| ResNet-50 | | | | | |
|---|---|---|---|---|---|
| Method | Accuracy (%) | Entropy ↑ | ECE (%) ↓ | AECE (%) ↓ | SCE ($\times 10^{-2}$) ↓ |
| Uncalibrated | 85.79 | 0.2256 | 5.77 | 5.76 | 1.7003 |
| Temperature Scaling | 85.79 ±0 | 0.3726 ±0 | 1.77 ±0 | 1.66 ±0 | 1.1067 ±0 |
| TS by Grid Search | 85.79 ±0 | 0.3684 ±0 | 2.13 ±0 | 1.68 ±0 | 1.1041 ±0 |
| Vector Scaling | 85.79 ±0.05 | 0.3653 ±0.0023 | 1.97 ±0.11 | 1.94 ±0.06 | 0.9264 ±0.0574 |
| Matrix Scaling | 85.79 ±0.38 | 0.2984 ±0.0161 | 4.96 ±0.65 | 4.86 ±0.71 | 1.4665 ±0.1314 |
| MS-ODIR | 85.79 ±0.04 | 0.3726 ±0.0001 | 1.94 ±0.01 | 1.70 ±0.03 | 0.9099 ±0.0101 |
| Dir-ODIR | 85.79 ±0.02 | 0.3748 ±0.0002 | 1.55 ±0.04 | 1.71 ±0.09 | 0.8366 ±0.0034 |
| Neural Clamping (CE) | 85.79 ±0.02 | 0.3820 ±0.0005 | 1.54 ±0.02 | 1.57 ±0.05 | 1.1100 ±0.0103 |
| Neural Clamping (FL) | 85.82 ±0.03 | 0.4204 ±0.0004 | 1.05 ±0.03 | 1.19 ±0.06 | 1.0797 ±0.0042 |

## 4 Performance Evaluation

In this section, we conducted extensive experiments to evaluate the performance of our proposed Neural Clamping calibration method using the calibration metrics introduced in Section 2.2. We compared our method to several baseline and state-of-the-art calibration methods. All experiments are evaluated on three popular image recognition datasets (BloodMNIST, CIFAR100, ImageNet-1K) and six trained deep neural network models (e.g. ResNet, Vision Transformer (ViT), and MLP-Mixer). An ablation study on Neural Clamping is presented at the end of this section.

### 4.1 Evaluation and Implementation Details

**Experiment setup.** We used ResNet-50 (He et al., 2016) on BloodMNIST (Yang et al., 2023); ResNet-110 (He et al., 2016) and Wide-ResNet-40-10 (Zagoruyko & Komodakis, 2016) models on CIFAR-100 (Krizhevsky et al., 2009); ResNet-101 (He et al., 2016), ViT-S/16 (Dosovitskiy et al., 2020), and MLP-Mixer B/16 (Tolstikhin et al., 2021) models on ImageNet-1K (Deng et al., 2009). Blood MNIST, a recognized medical machine learning benchmark, features 11,959/1,712/3,421 samples for training/validation/evaluation. For CIFAR-100 and ImageNet, lacking default validation data, we divided CIFAR-100's training set into 45,000 training images and 5,000 calibration images. For ImageNet, 25,000 test images were reserved for calibration,

Table 2: Comparison with various calibration methods on CIFAR-100 with different models. The reported results are mean and standard deviation over 5 runs. The best/second-best method is highlighted by blue/green color. On ECE, the relative improvement of Neural Clamping to the best baseline is 34/5 % on ResNet-110/Wide ResNet-40-10, respectively.

| ResNet-110 | | | | | |
|---|---|---|---|---|---|
| Method | Accuracy (%) | Entropy ↑ | ECE (%) ↓ | AECE (%) ↓ | SCE ($\times 10^{-2}$) ↓ |
| Uncalibrated | 74.15 | 0.4742 | 10.74 | 10.71 | 0.2763 |
| Temperature Scaling | 74.15 ±0 | 0.8991 ±0 | 1.71 ±0 | 1.63 ±0 | 0.1711 ±0 |
| TS by Grid Search | 74.15 ±0 | 0.9239 ±0 | 1.35 ±0 | 1.38 ±0 | 0.1717 ±0 |
| Vector Scaling | 73.81 ±0.05 | 0.8698 ±0.0008 | 2.29 ±0.07 | 2.15 ±0.15 | 0.1949 ±0.0046 |
| Matrix Scaling | 62.03 ±0.31 | 0.1552 ±0.0026 | 31.85 ±0.29 | 31.85 ±0.29 | 0.6842 ±0.0057 |
| MS-ODIR | 74.07 ±0.03 | 0.9035 ±0.0001 | 1.79 ±0.04 | 1.75 ±0.03 | 0.1797 ±0.0006 |
| Dir-ODIR | 74.10 ±0.04 | 0.9160 ±0.0002 | 1.36 ±0.05 | 1.31 ±0.03 | 0.1780 ±0.0014 |
| Neural Clamping (CE) | 74.17 ±0.07 | 0.8928 ±0.0061 | 1.67 ±0.16 | 1.63 ±0.19 | 0.1709 ±0.0020 |
| Neural Clamping (FL) | 74.16 ±0.09 | 0.9707 ±0.0049 | 0.89 ±0.06 | 1.01 ±0.11 | 0.1754 ±0.0015 |
| Wide-ResNet-40-10 | | | | | |
| Method | Accuracy (%) | Entropy ↑ | ECE (%) ↓ | AECE (%) ↓ | SCE ($\times 10^{-2}$) ↓ |
| Uncalibrated | 79.51 | 0.4210 | 7.63 | 7.63 | 0.2188 |
| Temperature Scaling | 79.51 ±0 | 0.7420 ±0 | 2.30 ±0 | 2.17 ±0 | 0.1627 ±0 |
| TS by Grid Search | 79.51 ±0 | 0.8359 ±0 | 1.75 ±0 | 1.54 ±0 | 0.1659 ±0 |
| Vector Scaling | 79.08 ±0.09 | 0.7079 ±0.0012 | 2.52 ±0.07 | 2.35 ±0.05 | 0.1818 ±0.0032 |
| Matrix Scaling | 68.48 ±0.16 | 0.1371 ±0.0023 | 26.13 ±0.15 | 26.12 ±0.15 | 0.5657 ±0.0024 |
| MS-ODIR | 79.15 ±0.03 | 0.7529 ±0.0002 | 1.90 ±0.07 | 1.95 ±0.03 | 0.1705 ±0.0008 |
| Dir-ODIR | 79.51 ±0.01 | 0.7707 ±0.0001 | 1.81 ±0.03 | 1.98 ±0.01 | 0.1625 ±0.0004 |
| Neural Clamping (CE) | 79.53 ±0.01 | 0.7461 ±0.0030 | 2.27 ±0.03 | 2.20 ±0.03 | 0.1624 ±0.0004 |
| Neural Clamping (FL) | 79.53 ±0.04 | 0.8626 ±0.0033 | 1.67 ±0.14 | 1.66 ±0.12 | 0.1683 ±0.0014 |

and the remaining 25,000 were for evaluation. Uniform calibration dataset and test set were shared across all methods. Our experiments ran on an Nvidia Tesla V100 with 32GB RAM and Intel Xeon Gold CPU.

**Comparative methods.** We compared our method to all the post-processing calibration methods introduced in Section 2.3, including Temperature Scaling (Guo et al., 2017), Vector Scaling, Matrix Scaling, Matrix scaling with ODIR (MS-ODIR) (Kull et al., 2019), and Dirichlet Calibration (Dir-ODIR) (Kull et al., 2019). For temperature scaling, we considered two implementations: (a) learning the temperature by minimizing NLL loss via gradient decent on the calibration dataset, and (b) taking a grid search on temperature over 0 to 5 with a resolution of 0.001 and then reporting the lowest ECE and its corresponding temperature, for which we call TS (Grid Searched). For MS-ODIR and Dir-ODIR, we trained their regularization coefficient with 7 values from $10^{-2}$ to $10^4$ and chose the best result on the calibration dataset (See Section 4.1 in (Kull et al., 2019)). All methods were trained with 1000 epochs with full-batch gradient descent with learning rate 0.001. In addition to Neural Clamping with the focal loss (FL), we also compared Neural Clamping with the cross entropy (CE) loss.

**Neural Clamping implementation.** The hyperparameters $\lambda$ and $\gamma$ in equation (9) are determined by the best parameter minimizing the ECE on the calibration dataset. The choice of $\gamma$ was already discussed in Sec 3.3. The default value of $\gamma$ is set to 1 because we find it to be stable across models and datasets. Regarding the choice of $\lambda$, its purpose is to aid in regularization. Our default approach is to set this term to be 1/10 of the initial loss. The input calibration parameter $\delta$ and the output calibration parameter $T$ are optimized using the stochastic gradient descent (SGD) optimizer with learning rate 0.001, batch size 512, and 100 epochs. For initialization, $\delta$ uses random initialization and $T$ is set to 1. The detailed algorithmic procedure of Neural Clamping is presented in Appendix A.

**Evaluation metrics.** We reported 5 evaluation measures on the test sets: Accuracy, Entropy, ECE, AECE, and SCE. All three calibration metrics are defined in Section 2.2 and 15 bins were used. In all experiments, we report the average value and standard deviation over 5 independent runs.

Table 3: Comparison with various calibration methods on ImageNet with different models. The reported results are mean and standard deviation over 5 runs. The best/second-best method is highlighted by blue/green color. On ECE, the relative improvement of Neural Clamping to the best baseline is 11/6/13 % on ResNet-101/ViT-S16/MLP-Mixer B16, respectively.

| ResNet-101 | | | | | |
|---|---|---|---|---|---|
| Method | Accuracy (%) | Entropy ↑ | ECE (%) ↓ | AECE (%) ↓ | SCE ($\times 10^{-3}$) ↓ |
| Uncalibrated | 75.73 | 0.6608 | 5.88 | 5.88 | 0.3180 |
| Temperature Scaling | 75.73 ±0 | 0.9376 ±0 | 1.88 ±0 | 1.91 ±0 | 0.3117 ±0 |
| TS by Grid Search | 75.73 ±0 | 0.9244 ±0 | 2.02 ±0 | 1.97 ±0 | 0.3108 ±0 |
| Vector Scaling | 75.67 ±0.07 | 1.0463 ±0.0017 | 2.04 ±0.12 | 1.92 ±0.07 | 0.3192 ±0.0009 |
| Matrix Scaling | 51.97 ±0.30 | 0.0593 ±0.0008 | 45.61 ±0.28 | 45.60 ±0.28 | 0.9037 ±0.0052 |
| MS-ODIR | 70.71 ±0.10 | 0.9904 ±0.0016 | 3.29 ±0.06 | 3.28 ±0.06 | 0.3448 ±0.0011 |
| Dir-ODIR | 70.72 ±0.03 | 0.9841 ±0.0007 | 3.47 ±0.05 | 3.47 ±0.05 | 0.3480 ±0.0013 |
| Neural Clamping (CE) | 75.73 ±0.01 | 0.9429 ±0.0240 | 1.89 ±0.13 | 1.88 ±0.11 | 0.3114 ±0.0007 |
| Neural Clamping (FL) | 75.73 ±0.01 | 1.0103 ±0.0245 | 1.68 ±0.04 | 1.71 ±0.03 | 0.3128 ±0.0001 |
| ViT-S/16 | | | | | |
| Method | Accuracy (%) | Entropy ↑ | ECE (%) ↓ | AECE (%) ↓ | SCE ($\times 10^{-3}$) ↓ |
| Uncalibrated | 79.90 | 0.7161 | 1.28 | 1.30 | 0.2808 |
| Temperature Scaling | 79.90 ±0 | 0.7314 ±0 | 1.08 ±0 | 1.09 ±0 | 0.2817 ±0 |
| TS by Grid Search | 79.90 ±0 | 0.7791 ±0 | 0.82 ±0 | 0.80 ±0 | 0.2852 ±0 |
| Vector Scaling | 80.02 ±0.03 | 0.9410 ±0.0014 | 2.62 ±0.02 | 2.69 ±0.03 | 0.2985 ±0.0015 |
| Matrix Scaling | 53.99 ±0.29 | 0.0646 ±0.0010 | 43.36 ±0.30 | 43.36 ±0.29 | 0.8811 ±0.0054 |
| MS-ODIR | 75.94 ±0.09 | 0.9810 ±0.0018 | 0.87 ±0.10 | 0.92 ±0.10 | 0.3163 ±0.0023 |
| Dir-ODIR | 75.93 ±0.09 | 0.9788 ±0.0007 | 0.93 ±0.06 | 0.86 ±0.09 | 0.3149 ±0.0018 |
| Neural Clamping (CE) | 79.98 ±0.01 | 0.7898 ±0.0028 | 0.81 ±0.03 | 0.77 ±0.04 | 0.2801 ±0.0005 |
| Neural Clamping (FL) | 79.97 ±0.01 | 0.7934 ±0.0038 | 0.77 ±0.01 | 0.72 ±0.03 | 0.2804 ±0.0004 |
| MLP-Mixer B/16 | | | | | |
| Method | Accuracy (%) | Entropy ↑ | ECE (%) ↓ | AECE (%) ↓ | SCE ($\times 10^{-3}$) ↓ |
| Uncalibrated | 73.94 | 0.6812 | 11.55 | 11.55 | 0.3589 |
| Temperature Scaling | 73.94 ±0 | 1.2735 ±0 | 4.94 ±0 | 4.98 ±0 | 0.3188 ±0 |
| TS by Grid Search | 73.94 ±0 | 1.6243 ±0 | 2.60 ±0 | 2.60 ±0 | 0.3258 ±0 |
| Vector Scaling | 73.24 ±0.06 | 1.1474 ±0.0089 | 6.91 ±0.17 | 6.88 ±0.20 | 0.3321 ±0.0027 |
| Matrix Scaling | 40.96 ±0.31 | 0.1137 ±0.0010 | 54.50 ±0.28 | 54.50 ±0.28 | 1.0979 ±0.0041 |
| MS-ODIR | 73.16 ±0.02 | 1.8049 ±0.0016 | 4.65 ±0.08 | 4.73 ±0.05 | 0.3477 ±0.0018 |
| Dir-ODIR | 73.13 ±0.05 | 1.8083 ±0.0013 | 4.68 ±0.09 | 4.76 ±0.09 | 0.3480 ±0.0018 |
| Neural Clamping (CE) | 74.14 ±0.01 | 1.7952 ±0.0302 | 2.43 ±0.16 | 2.51 ±0.18 | 0.3054 ±0.0020 |
| Neural Clamping (FL) | 74.12 ±0.00 | 1.7673 ±0.0269 | 2.27 ±0.13 | 2.34 ±0.14 | 0.3029 ±0.0018 |

Table 4: Ablation study with ResNet-110 on CIFAR-100. The best result is highlighted by blue color.

| Method | Accuracy (%) | Entropy ↑ | ECE (%) ↓ | AECE (%) ↓ | SCE ($\times 10^{-2}$) ↓ |
|---|---|---|---|---|---|
| Uncalibrated | 74.15 | 0.4742 | 10.74 | 10.71 | 0.2763 |
| Temperature Scaling | 74.15 | 0.8991 | 1.71 | 1.63 | 0.1711 |
| Temperature Scaling (FL) | 74.15 | 1.0542 | 2.01 | 2.02 | 0.1812 |
| Input Calibration w/ $\delta^*$ | 74.16 ±0.09 | 0.4775 ±0.03 | 10.62 ±0.10 | 10.61 ±0.10 | 0.2776 ±0.0019 |
| Output Calibration w/ $T^*$ | 74.15 ±0 | 0.9648 ±0.46 | 1.18 ±0.05 | 1.35 ±0.01 | 0.1731 ±0.0003 |
| Neural Clamping | 74.16 ±0.09 | 0.9707 ±0.49 | 0.89 ±0.06 | 1.01 ±0.11 | 0.1754 ±0.0015 |

## 4.2 BloodMNIST, CIFAR-100, and ImageNet Results

**BloodMNIST.** BloodMNIST is an 8-class microscopic peripheral blood cell image recognition task. The calibration results with ResNet-50 are shown in Table 1. Compared to the best existing method, Neural Clamping shows an additional 31%/28% reduction in ECE/AECE.

**CIFAR-100.** The experimental results on CIFAR-100 are presented in Table 2, which is divided into two sections corresponding to different models: ResNet-110 and Wide-ResNet-40-10. Our method consistently achieves the lowest ECE and either the lowest or second lowest AECE and SCE when compared to other existing methods. Notably, in the ResNet-110 experiment, Neural Clamping reduced ECE and AECE by 34% and 23%, respectively, compared to the best existing method. It is important to highlight that our method not only reduces calibration error but also improves accuracy, which sets it apart from existing approaches.

**ImageNet-1K.** Table 3 presents the experimental results on ImageNet, where the table is divided into three sections containing ResNet-101, ViT-S/16, and MLP-Mixer B/6. Neural Clamping consistently outperforms the compared methods by achieving the lowest ECE, AECE, and either the lowest or second-lowest SCE in all cases, similar to the CIFAR-100 experiments. Particularly, in the ResNet-101 experiment, Neural Clamping reduces ECE and AECE by 11% compared to the best existing method. Moreover, our method concurrently improves accuracy while reducing the calibration error for all three models, demonstrating its effectiveness in calibration for various model architectures.

Additionally, We also provide experimental results of different bins number in Appendix D, which clearly demonstrates the same conclusion of the outstanding calibration performance of Neural Clamping over the baselines. To further compare how our method differs from the baselines, we also visualize the ECE results via plotting the reliability diagrams in Appendix E.

### 4.3 Additional Analysis of Neural Clamping

**Data-driven vs. random initialization for input perturbation $\delta$.** There are two initialization methods for the input calibration $\delta$ in Neural Clamping: data-driven initialization as derived from Theorem 3.2 and random initialization. In scrutinizing these two initialization methods, we found that the data-driven initialization managed to consistently deliver stable calibration results. Figure 3 shows that across all metrics, both initialization methods have similar mean values across 5 runs, while the data-driven initialization has a smaller variation and standard deviation. As a result of the experiment, it can be concluded that data-driven value can not only offer a more reliable solution but also slightly improved outcomes. We also used this data-driven initialization to devise a computationally efficient Neural Clamping variant. Under similar runtime constraints, our method achieves superior calibration performance compared to temperature scaling. This implies that theoretically derived input perturbations can attain performance comparable to that of training results. Please see Appendix F for details.

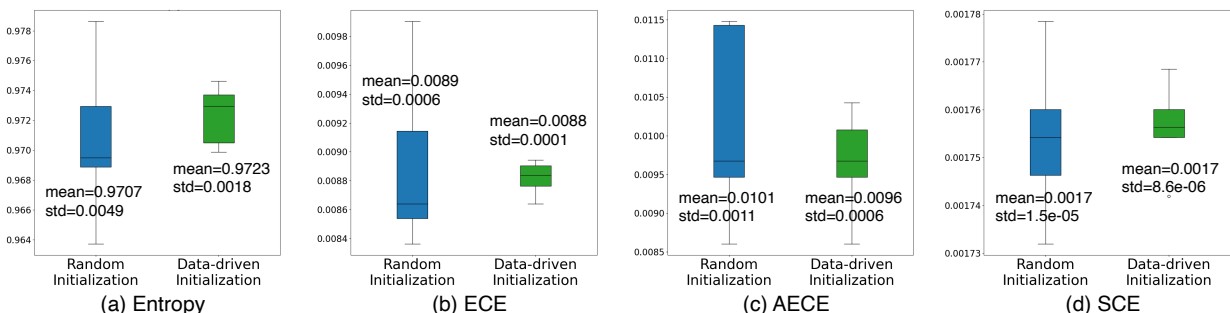

Figure 3: Comparison of random (blue) and data-driven (green) initializations for input calibration $\delta$ in Neural Clamping. The reported results are (a) Entropy, (b) ECE, (c) AECE, and (d) SCE of ResNet-110 on CIFAR-100 over 5 runs. This boxplot graphically demonstrates the spread groups of numerical data through their quartiles. The data-driven initialization shows better stability (smaller variation) than random initialization.

**Ablation study with input calibration $\delta^*$ and output calibration $T^*$.** After calibration, Neural Clamping learns $\delta^*$ for input calibration and $T^*$ for output calibration. In Table 4 we performed an ablation study to examine the effects of input calibration and output calibration separately with their jointly trained parameters $\delta^*$ and $T^*$ and "Temperature Scaling (FL)" approach, where $T$ optimized with focal loss.

For input calibration, we inferred testing data with only the learned input perturbation $\boldsymbol{\delta^*}$; for output calibration, we tested the result with only the learned temperature scaling parameter $T^*$. One noteworthy finding from this exercise is that while output calibration alone already trims the ECE and AECE materially, a further 25% reduction in ECE and AECE can be achieved when it is paired with input calibration (i.e. Neural Clamping). Input calibration alone is less effective because it does not directly modify the prediction output.

In addition, the Temperature Scaling (FL) actually performed worse than both the original Temperature Scaling method and "Output Calibration w/ T*" across the various calibration metrics. This finding suggests that the improvements achieved by our method Neural Clamping (joint input-output calibration) is not simply due to the use of temperature scaling with focal loss. This ablation study corroborates the necessity and advantage of joint input-output calibration and the additional benefits only gained from joint calibration framework.

## 5 Conclusion

In this paper, we present a new post-processing calibration method called Neural Clamping, which offers novel insights into joint input-output calibration and significantly improves calibration performance. We also develop theoretical analysis to justify the advantage of Neural Clamping. Our empirical results on several datasets and models show that Neural Clamping outperforms state-of-the-art post-processing calibration methods. We believe our method delivers a practical tool that can contribute to neural network based technology and applications requiring accurate calibration.

### Impact Statements

We see no ethical or immediate negative societal consequence of our work, and it holds the potential for positive social impacts. By improving the accuracy of machine learning models' prediction probabilities, our research can benefit various domains.

### Acknowledgments

I would like to express my gratitude to Yu-Chieh Cheng for his invaluable assistance in proofreading and editing.

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
