# Appendix

# A   Algorithmic Descriptions of Neural Clamping

---
**Algorithm 1** Neural Clamping
---
1: **Input:** Fixed $K$-way image classifier $f_\theta(\cdot)$, calibration dataset $\{\boldsymbol{x_i}, y_i\}_{i=1}^n$, learning rate $\epsilon$, focal loss hyperparameter $\gamma$, and weight-decay regularization hyperparameter $\lambda$
2: **Output:** The optimal input perturbation $\boldsymbol{\delta}$ and temperature $T$
3: **Initialize**
    $\boldsymbol{\delta} \leftarrow$ initialization (random or data-driven)
    $T \leftarrow 1$
    $Loss \leftarrow \mathcal{L}_{FL}^\gamma$ and $\lambda$ according to equation 9
4: **while** not converged **do**
5:     Sample data batch batches$(\boldsymbol{x_i}, y_i) \sim \{\boldsymbol{x_i}, y_i\}_{i=1}^n$
6:     **for** batches$(x_i, y_i)$ **do**
7:         Update $\boldsymbol{\delta} \leftarrow \boldsymbol{\delta} - \epsilon \nabla_{\boldsymbol{\delta}} Loss(f_\theta(\boldsymbol{x_i} + \boldsymbol{\delta})/T, y_i)$
8:         Update $T \leftarrow T - \epsilon \nabla_T Loss(f_\theta(\boldsymbol{x_i} + \boldsymbol{\delta})/T, y_i)$
9:     **end for**
10: **end while**
11: **return** $\boldsymbol{\delta}, T$
---

In our implementation, we set the hyperparameters $\lambda$ and $\gamma$ in equation (9) by the best parameter minimizing the ECE on the calibration dataset. The selection of $\lambda/\gamma$ sweeps from 0.001 to 10 and 0.01 to 5. with an increment of 0.001/0.01, respectively.

The input calibration parameter $\delta$ and the output calibration parameter $T$ are optimized using the stochastic gradient descent (SGD) optimizer with learning rate 0.001, batch size 512, and 100 epochs. For initialization, $\delta$ use randomly initialized (Gaussian distribution with mean=0 and variance=0.01) and $T$ is set to 1.

# B   Proof for Lemma 3.1

**Lemma 3.1** (optimality of joint input-output calibration)  *For any input perturbation $\boldsymbol{\delta}$, let $f_\theta(\cdot) = [f_\theta^{(1)}, \ldots, f_\theta^{(K)}]$ be a fixed $K$-way neural network classifier and let $\boldsymbol{z}$ be the output logits of a perturbed data input $\boldsymbol{x} + \boldsymbol{\delta}$. Then the proposed form of joint input-output calibration in Neural Clamping is the unique solution $q^*(\boldsymbol{z})^{(k)} = \frac{\exp[f_\theta^{(k)}(\boldsymbol{x}+\boldsymbol{\delta})/T]}{\sum_{j=1}^K \exp[f_\theta^{(j)}(\boldsymbol{x}+\boldsymbol{\delta})/T]}, \forall k \in \{1, \ldots, K\}$, to the constrained entropy maximization problem in equation 7.*

*Proof.* Without loss of generality, the following proof assumes a vectorized input dimension. Our proof extends the theoretical analysis on temperature scaling in the supplementary materials S.2 of Guo et al. (2017) to consider an input perturbation $\boldsymbol{\delta}$. We use the method of Lagrange multipliers to solve the constrained entropy maximization problem in equation 7. Let $\lambda_0 \in \mathbb{R}$ and $\lambda_1, \lambda_2, ..., \lambda_n \in \mathbb{R}$ be the Lagrangian multipliers for the constraint $\sum_{i=1}^n \boldsymbol{z_i}^\top \boldsymbol{e}^{(y_i)} = \sum_{i=1}^n \boldsymbol{z_i}^\top q(\boldsymbol{z_i})$ and $\sum_{i=1}^n \boldsymbol{1}^\top q(\boldsymbol{z_i}) = 1$, $\forall i$, respectively. We will show the optimal solution automatically satisfies the first constraint (nonnegativity) $q(\boldsymbol{z_i})^{(k)} \geq 0$ for all $i$ and $k$ later. Then we define

$$L = -\sum_{i=1}^n q(\boldsymbol{z_i})^\top \log(q(\boldsymbol{z_i})) + \lambda_0 \sum_{i=1}^n \left[ \boldsymbol{z_i}^\top q(\boldsymbol{z_i}) - \boldsymbol{z_i} \cdot \boldsymbol{e}^{(y_i)} \right]$$
$$+ \sum_{i=1}^n \lambda_i \left[ \boldsymbol{1}^\top q(\boldsymbol{z_i}) - 1 \right] \tag{10}$$

Taking the partial derivative of $L$ with respect to $q(z_i)$ gives

$$\frac{\partial}{\partial q(z_i)} L = -log(q(\boldsymbol{z_i})) - \boldsymbol{1} + \lambda_0 \boldsymbol{z_i} + \lambda_i \boldsymbol{1} \tag{11}$$

Let the partial derivative of $L$ equal 0, then we can get

$$q(\boldsymbol{z_i}) = \begin{bmatrix} \exp[\lambda_0 z_i^{(1)} + \lambda_i - 1] \\ \vdots \\ \exp[\lambda_0 z_i^{(K)} + \lambda_i - 1] \end{bmatrix} \tag{12}$$

Note that this expression suggests $q(\boldsymbol{z_i})^{(k)} \geq 0$ and thus the first constraint is satisfied. Due to the constraint $\sum_{i=1}^{n} \mathbf{1}^\top q(\boldsymbol{z_i}) = 1$ for all $i$, the solution $q(z_i)$ must be

$$q(z_i)^{(k)} = \frac{\exp[\lambda_0 z_i^{(k)}]}{\sum_{j=1}^{K} \exp[\lambda_0 z_i^{(j)}]} \tag{13}$$

By setting $T = \frac{1}{\lambda_0}$, we can get the unique solution

$$q^*(\boldsymbol{z})^{(k)} = \frac{\exp[f_\theta^{(k)}(\boldsymbol{x} + \boldsymbol{\delta})/T]}{\sum_{j=1}^{K} \exp[f_\theta^{(j)}(\boldsymbol{x} + \boldsymbol{\delta})/T]}, \ \forall k \in \{1, \ldots, K\}. \tag{14}$$

$\square$

## C   Proof for Theorem 3.2

**Theorem 3.2** (provable entropy increment and data-driven initialization)  *Let $[\boldsymbol{\alpha}, \boldsymbol{\beta}]$ be the feasible range of data inputs and $\boldsymbol{g} = \sum_{i=1}^{n} g_i = [g^{(1)}, \ldots, g^{(K)}]$ be the sum of local input gradients. Define $\boldsymbol{\eta} \in \mathbb{R}^m$ element-wise such that $\eta_j = \ell_j - \alpha_j$ if $g^{(j)} < 0$, $\eta_j = \beta_j - \mu_j$ if $g^{(j)} > 0$, and $\eta_j = 0$ otherwise, for every $j \in \{1, \ldots, m\}$. Approaching by first-order approximation and given the same temperature value $T$, Neural Clamping increases the entropy of temperature scaling by $\boldsymbol{\delta}^\top \boldsymbol{g}$. Furthermore, the optimal value $\widetilde{\boldsymbol{\delta}}$ for maximizing $\boldsymbol{\delta}^\top \boldsymbol{g}$ is $\widetilde{\boldsymbol{\delta}} = \mathsf{sign}(\boldsymbol{g}) \odot \boldsymbol{\eta}$.*

*Proof.* For ease of understanding, let $\hat{H}(\boldsymbol{x}) = H(\sigma(f_\theta(\boldsymbol{x})/T))$ denote the entropy of the classifier $f_\theta$ (with softmax as the final output layer) after calibration. We have Taylor series expansion of $\hat{H}$ at a point $x_0$ as:

$$\begin{aligned} \hat{H}(\boldsymbol{x}) = \hat{H}(\boldsymbol{x_0}) &+ (\boldsymbol{x} - \boldsymbol{x_0})^\top \nabla \hat{H}(\boldsymbol{x_0}) \\ &+ \frac{1}{2}(\boldsymbol{x} - \boldsymbol{x_0})^\top \nabla^2 \hat{H}(\boldsymbol{x_0})(\boldsymbol{x} - \boldsymbol{x_0}) + \cdots \end{aligned} \tag{15}$$

Adding input perturbation $\delta$ to input data point $x$ and applying the first-order approximation on $\hat{H}(x)$, we can get

$$\begin{aligned} \hat{H}(\boldsymbol{x} + \boldsymbol{\delta}) &= \hat{H}(\boldsymbol{x}) + [(\boldsymbol{x} + \boldsymbol{\delta}) - \boldsymbol{x}]^\top \nabla \hat{H}(\boldsymbol{x}) + \cdots \\ &\approx \hat{H}(\boldsymbol{x}) + \boldsymbol{\delta}^\top \nabla \hat{H}(\boldsymbol{x}) \end{aligned} \tag{16}$$

Then we can use above approximation to compute the average output entropy for all data $\{x_i\}_{i=1}^{n}$:

$$\frac{1}{n} \sum_{i=1}^{n} \hat{H}(\boldsymbol{x_i} + \boldsymbol{\delta}) = \frac{1}{n} \sum_{i=1}^{n} \hat{H}(\boldsymbol{x_i}) + \frac{1}{n} \sum_{i=1}^{n} \boldsymbol{\delta}^\top \nabla \hat{H}(\boldsymbol{x_i}) \tag{17}$$

Let $\boldsymbol{g_i} = \nabla \hat{H}(\boldsymbol{x_i}) = \nabla H(\sigma(f_\theta(\boldsymbol{x_i})/T))$ is the input gradient with respect to $\boldsymbol{x_i}$, and $\boldsymbol{g}$ is the average input gradient $\frac{1}{n}\sum_{i=1}^{n} \boldsymbol{g_i}$. The first term is the original entropy value, namely the entropy of temperature scaling. The second term is the additional entropy term caused by introducing the input perturbation. The latter can be rewritten as:

$$\triangle \hat{H}(\boldsymbol{x_i} + \boldsymbol{\delta}) = \frac{1}{n} \sum_{i=1}^{n} \boldsymbol{\delta}^\top \nabla \hat{H}(\boldsymbol{x_i}) = \frac{\boldsymbol{\delta}^\top}{n} \sum_{i=1}^{n} \boldsymbol{g_i} = \boldsymbol{\delta}^\top \boldsymbol{g} \tag{18}$$

Therefore, Neural Clamping increases the entropy of temperature scaling by $\boldsymbol{\delta}^\top \boldsymbol{g}$.

Maximizing the scalar product $\boldsymbol{\delta}^\top \boldsymbol{g}$ under the constraint $\boldsymbol{x_i} + \boldsymbol{\delta} \in [\boldsymbol{\alpha}, \boldsymbol{\beta}]$ for all $i$ is equivalent to maximizing an inner product over an $L_\infty$ ball, where $[\boldsymbol{\alpha}, \boldsymbol{\beta}] \subset \mathbb{R}^m \times \mathbb{R}^m$ means the bounded range of all feasible data inputs, e.g., every image pixel value is within $[0, 255]$.

Due to the constraint, we further define $\boldsymbol{\ell} \in \mathbb{R}^m$ and $\boldsymbol{\mu} \in \mathbb{R}^m$ as the lower bound and the upper bound over all calibration data $\{\boldsymbol{x_i}\}_{i=1}^n$ on each input dimension. That is, their $j$-th entry is defined as $\ell_j = \min_{i \in \{1,...,n\}} x_i^{(j)}$ and $\mu_j = \max_{i \in \{1,...,n\}} x_i^{(j)}$, respectively. Then we can find available range value $\boldsymbol{\eta} \in \mathbb{R}^m$ for $\boldsymbol{\delta}$ to maximize the scalar product $\boldsymbol{\delta}^\top \boldsymbol{g}$ according to the direction of input gradient, $\boldsymbol{\eta}$ can be defined as:

$$\eta_j = \begin{cases} \ell_j - \alpha_j & \text{if } g^{(j)} < 0 \\ \beta_j - \mu_j & \text{if } g^{(j)} > 0 \\ 0 & \text{otherwise} \end{cases} \tag{19}$$

Finally, we can get the optimal $\widetilde{\boldsymbol{\delta}}$ for maximizing the scalar product $\boldsymbol{\delta}^\top \boldsymbol{g}$, i.e.,

$$\widetilde{\boldsymbol{\delta}} = \mathsf{sign}(\boldsymbol{g}) \odot \boldsymbol{\eta} \tag{20}$$

$\square$

# D    Result with Varying Bin Numbers

Calibration error measurements are known to be influenced by the number of bins. In order to account for the influence of bin numbers on calibration error measurements, we present additional results using different bin configurations. Specifically, we evaluate the results of the BloodMNIST experiment with bin numbers 10 and 20, which are displayed in Table 5 and Table 6, respectively. Furthermore, we provide the results of the CIFAR-100 experiment with bin numbers 10 and 20 in Table 7 and Table 8, respectively. Lastly, the results of the ImageNet-1K experiment with bin numbers 10 and 20 can be found in Table 9 and Table 10, respectively. By examining the results across different bin configurations, we gain a more comprehensive understanding of the performance of our approach in different scenarios.

Table 5: Comparison with various calibration methods on BloodMNIST with ResNet-50 (calibration metric bins=10). The reported results are mean and standard deviation over 5 runs. The best/second-best method is highlighted by blue/green color. On ECE/AECE, the relative improvement of Neural Clamping to the best baseline is 21% and 28%, respectively.

| | ResNet-50 | | | | |
|---|---|---|---|---|---|
| Method | Accuracy (%) | Entropy $\uparrow$ | ECE (%) $\downarrow$ | AECE (%) $\downarrow$ | SCE ($\times 10^{-2}$) $\downarrow$ |
| Uncalibrated | 85.79 | 0.2256 | 5.77 | 5.76 | 1.6217 |
| Temp. Scaling | 85.79 $\pm 0$ | 0.3726 $\pm 0$ | 1.32 $\pm 0$ | 1.43 $\pm 0$ | 1.0148 $\pm 0$ |
| TS by Grid Search | 85.79 $\pm 0$ | 0.3726 $\pm 0$ | 1.86 $\pm 0$ | 1.59 $\pm 0$ | 1.0134 $\pm 0$ |
| Vector Scaling | 85.79 $\pm 0.05$ | 0.3653 $\pm 0.0023$ | 1.80 $\pm 0.09$ | 1.87 $\pm 0.17$ | 0.8355 $\pm 0.0688$ |
| Matrix Scaling | 85.79 $\pm 0.38$ | 0.2984 $\pm 0.0161$ | 4.93 $\pm 0.67$ | 4.85 $\pm 0.72$ | 1.3964 $\pm 0.1345$ |
| MS-ODIR | 85.79 $\pm 0.04$ | 0.3726 $\pm 0.0001$ | 1.59 $\pm 0.03$ | 1.65 $\pm 0.05$ | 0.7277 $\pm 0.0021$ |
| Dir-ODIR | 85.79 $\pm 0.02$ | 0.3748 $\pm 0.0002$ | 1.94 $\pm 0.07$ | 1.39 $\pm 0.04$ | 0.7435 $\pm 0.0141$ |
| NC (CE) | 85.79 $\pm 0.02$ | 0.3820 $\pm 0.0005$ | 1.20 $\pm 0.04$ | 1.32 $\pm 0.04$ | 1.0131 $\pm 0.0063$ |
| NC (FL) | 85.82 $\pm 0.03$ | 0.4204 $\pm 0.0004$ | 1.04 $\pm 0.04$ | 0.99 $\pm 0.04$ | 0.9744 $\pm 0.0031$ |

# E    Reliability Diagrams

To visually compare the ECE results (15 bins) to each method with the groundtruth, we present reliability diagrams. These diagrams offer a comprehensive view of the calibration performance. The reliability

Table 6: Comparison with various calibration methods on BloodMNIST with ResNet-50 (calibration metric bins=20). The reported results are mean and standard deviation over 5 runs. The best/second-best method is highlighted by blue/green color. On ECE/AECE, the relative improvement of Neural Clamping to the best baseline is 26% and 22%, respectively.

| ResNet-50 | | | | | |
|---|---|---|---|---|---|
| Method | Accuracy (%) | Entropy ↑ | ECE (%) ↓ | AECE (%) ↓ | SCE ($\times 10^{-2}$) ↓ |
| Uncalibrated | 85.79 | 0.2256 | 5.77 | 5.76 | 1.7247 |
| Temp. Scaling | 85.79 ±0 | 0.3726 ±0 | 1.62 ±0 | 1.49 ±0 | 1.1978 ±0 |
| TS by Grid Search | 85.79 ±0 | 0.3726 ±0 | 2.00 ±0 | 1.66 ±0 | 1.1644 ±0 |
| Vector Scaling | 85.79 ±0.05 | 0.3653 ±0.0023 | 2.11 ±0.15 | 2.09 ±0.10 | 1.0276 ±0.0798 |
| Matrix Scaling | 85.79 ±0.38 | 0.2984 ±0.0161 | 5.00 ±0.63 | 4.94 ±0.66 | 1.5581 ±0.1318 |
| MS-ODIR | 85.79 ±0.04 | 0.3726 ±0.0001 | 2.08 ±0.06 | 2.32 ±0.06 | 0.9422 ±0.0107 |
| Dir-ODIR | 85.79 ±0.02 | 0.3748 ±0.0002 | 2.05 ±0.07 | 1.50 ±0.02 | 0.9563 ±0.0206 |
| NC (CE) | 85.79 ±0.02 | 0.3820 ±0.0005 | 1.83 ±0.13 | 1.43 ±0.05 | 1.2076 ±0.0116 |
| NC (FL) | 85.82 ±0.03 | 0.4204 ±0.0004 | 1.19 ±0.08 | 1.17 ±0.06 | 1.1905 ±0.0031 |

diagrams of BloodMNIST, CIFAR-100, and ImageNet are presented in Figure 4, Figure 5, and Figure 6, respectively. Pink color is the perfectly calibrated, and purple color is the actual probability of the output. By examining these diagrams, we gain graphical insights into the effectiveness of each method's calibration performance.

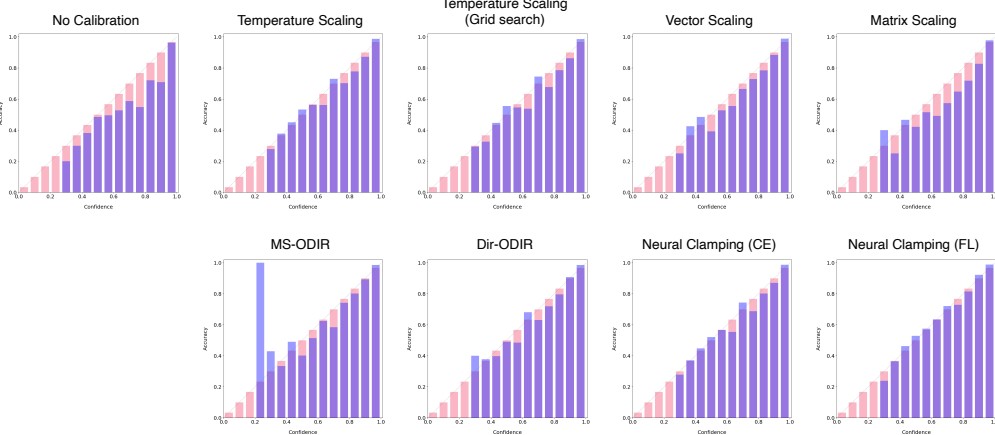

Figure 4: Reliability diagram of ResNet-50 on BloodMNIST with 15 bins ECE metric

## F Computationally Efficient Neural Clamping

We utilize our Theorem 1 to develop a Computationally Efficient Neural Clamping approach, referred to as NC (Eff.). NC (Eff.) adopts the data-driven initialization $\widetilde{\delta}$ for the input perturbation (input gradient w.r.t. entropy as discussed in Sec. 3.4), followed by temperature scaling (TS) with grid search (TS-Grid). This lightweight version spares the need for training input perturbation, requires only one additional backpropagation, and does not add additional hyperparameter tuning.

To demonstrate the effectiveness of NC (Eff.), we present two examples: ResNet-50 on BloodMNIST and ResNet-110 on CIFAR-100, in Table 11. We observed similar results across other examples as well. In our paper, the resolution for temperature scaling (grid search) was set to 0.001, and we included a comparison group with a resolution of 0.01 for analysis purposes. The table clearly shows that regardless of the resolution used, NC (Eff.) consistently outperforms temperature scaling (grid search) in terms of ECE, with

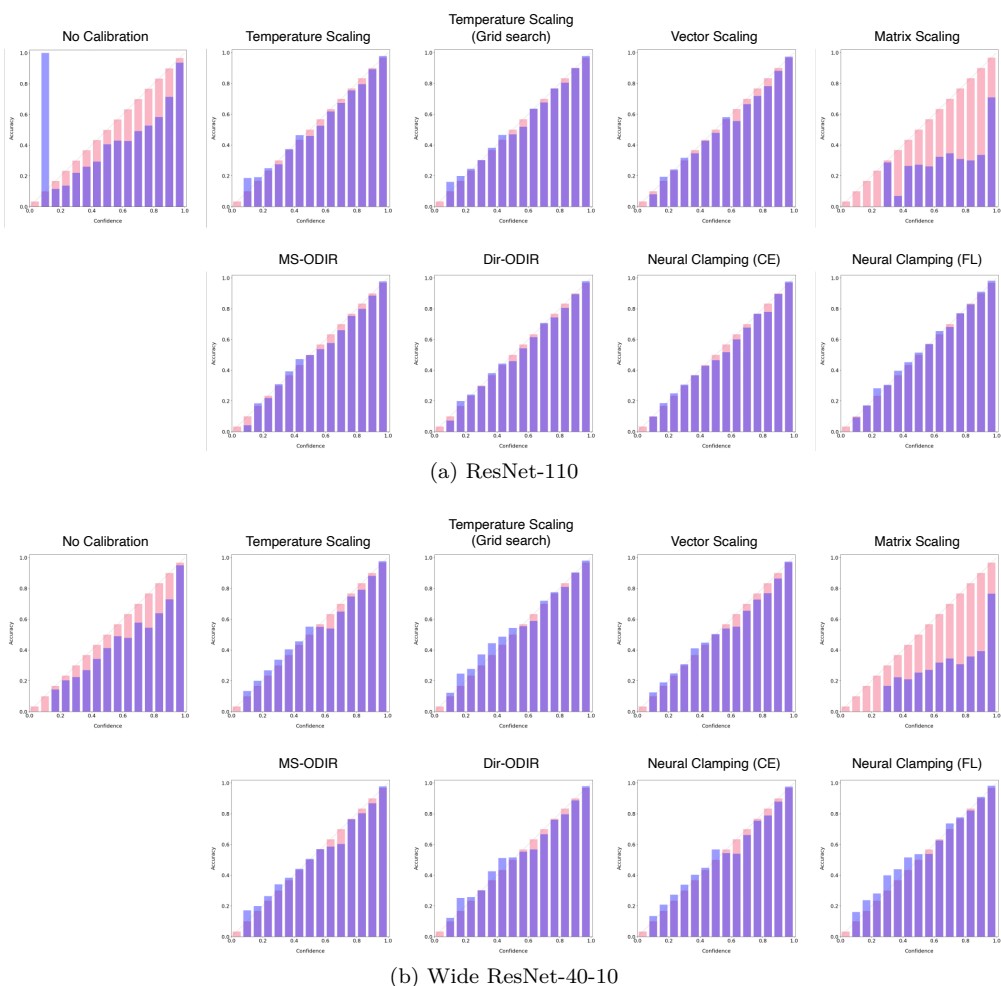

Figure 5: Reliability diagram of (a) ResNet-110 and (b) Wide ResNet-40-10 on CIFAR-100 with 15 bins ECE metric

improvements of up to 33.7 %. Notably, even with lower resolution, NC (Eff.) can still achieve better results than high-resolution temperature scaling in terms of both speed and ECE.

## G   Calibration Experiment on Model Trained with Different Initialization Seeds

We trained ResNet-110 on CIFAR-100 with 5 random initialization seeds to account for potential variability in the training process. For each of the 5 trained models, we performed the post-training calibration using a 5-fold cross-validation approach. This means that for each of the 5 trained models, we obtained 5 calibration datasets, resulting in a total of 25 calibration runs (5 models × 5 folds).

While the 25 calibration runs do not strictly adhere to the assumption of independent and identically distributed (IID) samples, the 5-fold cross-validation approach mitigates dependence and 5 random initializations introduce variation, enabling a robust comparison. The results are shown in Table 12. Then, we conducted a Welch's t-test to compare our proposed method (Neural Clamping) and the second-best baseline method (TS by Grid Search). The results show:

- p-value = 0.00000000000004

- t-statistic = 10.67

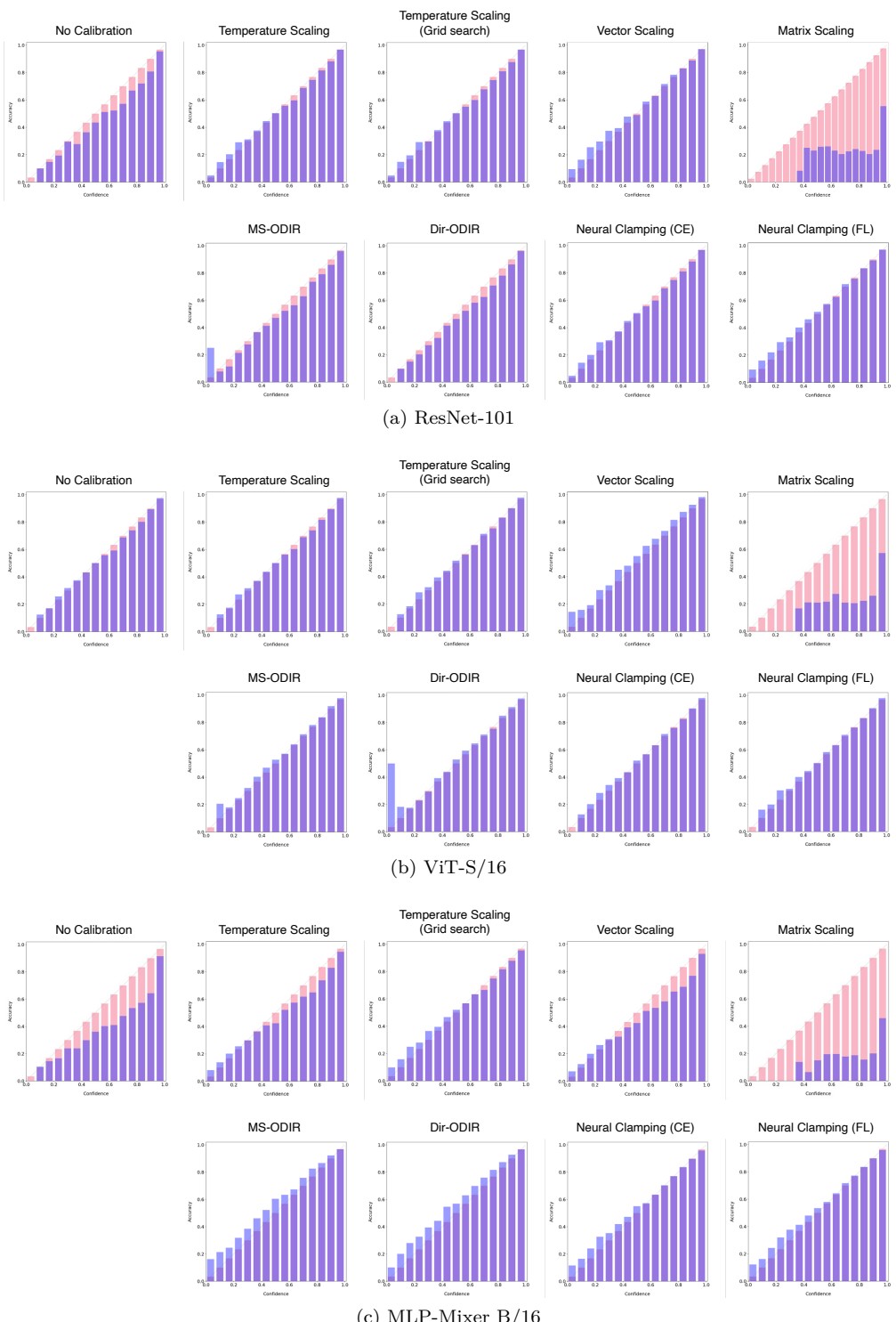

Figure 6: Reliability diagram of (a) ResNet-101, (b) ViT-S/16, and (c) MLP-Mixer B/16 on ImageNet with 15 bins ECE metric

The extremely small p-value (well below the 0.05 significance level) indicates that there is a highly statistically significant difference between the Neural Clamping method and the TS by Grid Search method. The absolute value of the t-statistic, 10.67, is far greater than the critical value of 2, further confirming the highly

Table 7: Comparison with various calibration methods on CIFAR-100 with different models (calibration metric bins=10). The reported results are mean and standard deviation over 5 runs. The best/second-best method is highlighted by blue/green color. On ECE, the relative improvement of Neural Clamping to the best baseline is 26/2 % on ResNet-110/Wide ResNet-40-10, respectively.

| ResNet-110 | | | | | |
|---|---|---|---|---|---|
| Method | Accuracy (%) | Entropy ↑ | ECE (%) ↓ | AECE (%) ↓ | SCE ($\times 10^{-2}$) ↓ |
| Uncalibrated | 74.15 | 0.47430 | 10.707 | 10.714 | 0.26065 |
| Temperature Scaling | 74.15 ±0 | 0.8991 ±0 | 1.37 ±0 | 1.48 ±0 | 0.1443 ±0 |
| TS by Grid Search | 74.15 ±0 | 0.9239 ±0 | 1.08 ±0 | 1.26 ±0 | 0.1425 ±0 |
| Vector Scaling | 73.81 ±0.05 | 0.8698 ±0.0008 | 2.16 ±0.14 | 2.13 ±0.18 | 0.1683 ±0.0031 |
| Matrix Scaling | 62.03 ±0.31 | 0.1552 ±0.0026 | 31.86 ±0.29 | 31.85 ±0.29 | 0.6749 ±0.0060 |
| MS-ODIR | 74.07 ±0.03 | 0.9035 ±0.0001 | 1.67 ±0.04 | 1.79 ±0.03 | 0.1555 ±0.0009 |
| Dir-ODIR | 74.10 ±0.04 | 0.9160 ±0.0002 | 1.16 ±0.03 | 1.19 ±0.08 | 0.1501 ±0.0008 |
| Neural Clamping (CE) | 74.17 ±0.07 | 0.8928 ±0.0061 | 1.60 ±0.19 | 1.50 ±0.11 | 0.1427 ±0.0012 |
| Neural Clamping (FL) | 74.16 ±0.09 | 0.9707 ±0.0049 | 0.80 ±0.12 | 0.86 ±0.07 | 0.1486 ±0.0015 |
| Wide-ResNet-40-10 | | | | | |
| Method | Accuracy (%) | Entropy ↑ | ECE (%) ↓ | AECE (%) ↓ | SCE ($\times 10^{-2}$) ↓ |
| Uncalibrated | 79.51 | 0.4211 | 7.63 | 7.63 | 0.2009 |
| Temperature Scaling | 79.51 ±0 | 0.7421 ±0 | 2.17 ±0 | 2.18 ±0 | 0.1369 ±0 |
| TS by Grid Search | 79.51 ±0 | 0.8359 ±0 | 1.65 ±0 | 1.48 ±0 | 0.1417 ±0 |
| Vector Scaling | 79.08 ±0.09 | 0.7079 ±0.0012 | 2.49 ±0.08 | 2.33 ±0.07 | 0.1612 ±0.0033 |
| Matrix Scaling | 68.48 ±0.16 | 0.1372 ±0.0023 | 26.14 ±0.14 | 26.13 ±0.15 | 0.5563 ±0.0020 |
| MS-ODIR | 79.15 ±0.03 | 0.7529 ±0.0002 | 1.90 ±0.04 | 1.95 ±0.03 | 0.1501 ±0.0005 |
| Dir-ODIR | 79.51 ±0.02 | 0.7707 ±0.0001 | 1.74 ±0.02 | 1.98 ±0.01 | 0.1366 ±0.0007 |
| Neural Clamping (CE) | 79.53 ±0.01 | 0.7462 ±0.0030 | 2.15 ±0.06 | 2.22 ±0.03 | 0.1368 ±0.0003 |
| Neural Clamping (FL) | 79.53 ±0.04 | 0.8626 ±0.0033 | 1.61 ±0.10 | 1.61 ±0.10 | 0.1445 ±0.0008 |

significant difference between the two methods. In summary, experimental results demonstrate that the Neural Clamping method has a significantly superior calibration performance compared to other baseline methods.

# H Statistical Significance Tests

We check all experiments in our main paper, our proposed method (Neural Clamping) shows extremely statistically significant differences to the corresponding second-best baseline method (such as Dir-ODIR, TS(GS), TS, etc.). The p-values are extremely low, typically less than 0.001 significance level, and some even less than 0.00001. This means the observed differences are highly unlikely to have occurred by random chance. The absolute values of the t-statistics are very high, often above 10, and some even exceeding 80. This indicates the performance gap between the two methods is very large. These statistical metrics provide very strong evidence that your proposed Neural Clamping (FL) method significantly outperforms the baseline methods across various datasets and network architectures.

- ResNet-50 @ BloodMNIST Neural Clamping (FL)) vs. Dir-ODIR:
  t-statistic: 22.36 and p-value: 0.000000044

- ResNet-110 @ CIFIAR-100 Neural Clamping (FL) vs. TS(GS):
  t-statistic: 17.14 and p-value: 0.000067

- Wide-ResNet-40-10 @ CIFAR-100 Neural Clamping (FL) vs. TS (GS):
  t-statistic: 23.88 and p-value: 0.000018

- ResNet-101 @ ImageNet-1K Neural Clamping (FL) vs. TS:
  t-statistic: 83.41 and p-value: 0.00000012

Table 8: Comparison with various calibration methods on CIFAR-100 with different models (calibration metric bins=20). The reported results are mean and standard deviation over 5 runs. The best/second-best method is highlighted by blue/green color. On ECE, the relative improvement of Neural Clamping to the best baseline is 26/6 % on ResNet-110/Wide ResNet-40-10, respectively.

| ResNet-110 | | | | | |
|---|---|---|---|---|---|
| Method | Accuracy (%) | Entropy ↑ | ECE (%) ↓ | AECE (%) ↓ | SCE ($\times 10^{-2}$) ↓ |
| Uncalibrated | 74.15 | 0.4743 | 10.74 | 10.71 | 0.2937 |
| Temperature Scaling | 74.15 ±0 | 0.8991 ±0 | 1.72 ±0 | 1.68 ±0 | 0.1943 ±0 |
| TS by Grid Search | 74.15 ±0 | 0.9240 ±0 | 1.62 ±0 | 1.51 ±0 | 0.1938 ±0 |
| Vector Scaling | 73.81 ±0.05 | 0.8699 ±0.0008 | 2.31 ±0.19 | 2.25 ±0.21 | 0.2155 ±0.0027 |
| Matrix Scaling | 62.03 ±0.31 | 0.1552 ±0.0026 | 31.86 ±0.29 | 31.86 ±0.29 | 0.6914 ±0.0060 |
| MS-ODIR | 74.07 ±0.03 | 0.9035 ±0.0001 | 1.89 ±0.08 | 1.82 ±0.02 | 0.2031 ±0.0010 |
| Dir-ODIR | 74.10 ±0.04 | 0.9160 ±0.0002 | 1.48 ±0.07 | 1.44 ±0.20 | 0.2046 ±0.0020 |
| Neural Clamping (CE) | 74.17 ±0.07 | 0.8929 ±0.0061 | 1.78 ±0.18 | 1.62 ±0.12 | 0.1937 ±0.0029 |
| Neural Clamping (FL) | 74.16 ±0.09 | 0.9941 ±0.0049 | 1.09 ±0.16 | 1.20 ±0.17 | 0.2003 ±0.0029 |
| Wide-ResNet-40-10 | | | | | |
| Method | Accuracy (%) | Entropy ↑ | ECE (%) ↓ | AECE (%) ↓ | SCE ($\times 10^{-2}$) ↓ |
| Uncalibrated | 79.51 | 0.4211 | 7.63 | 7.63 | 0.2351 |
| Temperature Scaling | 79.51 ±0 | 0.7421 ±0 | 2.22 ±0 | 2.21 ±0 | 0.1818 ±0 |
| TS by Grid Search | 79.51 ±0 | 0.8359 ±0 | 1.74 ±0 | 1.75 ±0 | 0.1900 ±0 |
| Vector Scaling | 79.08 ±0.09 | 0.7079 ±0.0012 | 2.59 ±0.09 | 2.47 ±0.08 | 0.2020 ±0.0032 |
| Matrix Scaling | 68.48 ±0.16 | 0.1372 ±0.0023 | 26.14 ±0.14 | 26.13 ±0.15 | 0.5728 ±0.0013 |
| MS-ODIR | 79.15 ±0.03 | 0.7529 ±0.0002 | 1.95 ±0.04 | 1.96 ±0.03 | 0.1915 ±0.0006 |
| Dir-ODIR | 79.51 ±0.01 | 0.7707 ±0.0001 | 1.94 ±0.02 | 1.99 ±0.01 | 0.1834 ±0.0008 |
| Neural Clamping (CE) | 79.53 ±0.01 | 0.7462 ±0.0030 | 2.20 ±0.03 | 2.24 ±0.04 | 0.1816 ±0.0003 |
| Neural Clamping (FL) | 79.53 ±0.04 | 0.8626 ±0.0033 | 1.63 ±0.06 | 1.70 ±0.14 | 0.1916 ±0.0019 |

- ViT-S/16 @ ImageNet-1K Neural Clamping (FL) vs. TS (GS):
  t-statistic: 11.18 and p-value: 0.00036

- MLP-Mixer B/16 @ ImageNet-1K Neural Clamping (FL) vs. TS (GS):
  t-statistic: 5.68 and p-value: 0.0047

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

Table 11: Time comparison with Computationally Efficient Neural Clamping (NC (Eff.)) and temperature scaling by grid search.

| Method | Resolution | Acc. (%) | Entropy ↑ | ECE (%) ↓ | Time (s) |
|---|---|---|---|---|---|
| ResNet-50 on BloodMNIST | | | | | |
| Uncalibrated | N/A | 85.79 | 0.2256 | 5.77 | N/A |
| TS-Grid | 0.01 | 85.79 | 0.3654 | 2.16 | 1.5 |
| TS-Grid | 0.001 | 85.79 | 0.3684 | 2.13 | 11.9 |
| NC (Eff.) | 0.01 | 85.79 | 0.3953 | 1.47 | 5.8 |
| NC (Eff.) | 0.001 | 85.79 | 0.3953 | 1.43 | 16.2 |
| NC (FL) | N/A | 85.79 | 0.4204 | 1.05 | 35.0 |
| ResNet-110 on CIFAR-100 | | | | | |
| Uncalibrated | N/A | 74.15 | 0.4742 | 10.74 | N/A |
| TS-Grid | 0.01 | 74.15 | 0.9268 | 1.36 | 2.2 |
| TS-Grid | 0.001 | 74.15 | 0.9239 | 1.35 | 13.0 |
| NC (Eff.) | 0.01 | 74.19 | 0.9371 | 1.23 | 7.2 |
| NC (Eff.) | 0.001 | 74.19 | 0.9342 | 1.23 | 18.0 |
| NC (FL) | N/A | 74.16 | 0.9707 | 0.89 | 227.0 |

Table 12: Comparison with various calibration methods on CIFAR-100 with ResNet-110 with different training seeds (calibration metric bins=15). The reported results are mean and standard deviation over 25 runs (5 training initialization seeds × 5 calibration runs). The best/second-best method is highlighted by blue/green color. We conducted a Welch's t-test to compare our proposed method (Neural Clamping) and the second-best baseline method (TS by Grid Search), the results also showed highly significant difference between the two methods

| Method | Accuracy (%) | Entropy ↑ | ECE (%) ↓ | AECE (%) ↓ | SCE ($\times 10^{-2}$) ↓ |
|---|---|---|---|---|---|
| ResNet-110 on CIFAR (training with different seeds) | | | | | |
| Uncalibrated | 73.95 ±0.28 | 0.5041 ±0.0144 | 10.14 ±0.69 | 10.12 ±0.68 | 0.2722 ±0.1196 |
| Temperature Scaling | 73.95 ±0.28 | 0.9146 ±0.0169 | 1.75 ±0.68 | 1.56 ±0.90 | 0.1776 ±0.0304 |
| Temperature Scaling (FL) | 73.95 ±0.28 | 1.0777 ±0.0036 | 2.35 ±0.36 | 2.35 ±0.36 | 0.1889 ±0.0267 |
| TS by Grid Search | 73.95 ±0.28 | 0.9476 ±0.0040 | 1.49 ±0.19 | 1.27 ±0.18 | 0.1480 ±0.0434 |
| Vector Scaling | 73.45 ±0.78 | 0.8738 ±0.0164 | 2.27 ±0.21 | 2.11 ±0.22 | 0.1799 ±0.0449 |
| Matrix Scaling | 63.95 ±1.85 | 0.1648 ±0.0055 | 31.71 ±0.41 | 31.71 ±0.41 | 0.6813 ±0.0830 |
| MS-ODIR | 73.58 ±0.31 | 0.9187 ±0.0230 | 1.83 ±0.22 | 1.51 ±0.28 | 0.1801 ±0.0304 |
| Dir-ODIR | 73.78 ±0.32 | 0.9504 ±0.0140 | 1.58 ±0.24 | 1.32 ±0.21 | 0.1762 ±0.0290 |
| Neural Clamping (CE) | 74.16 ±0.47 | 0.9514 ±0.0314 | 1.06 ±0.15 | 1.14 ±0.17 | 0.1634 ±0.0506 |
| Neural Clamping (FL) | 74.17 ±0.61 | 0.9921 ±0.0289 | 0.96 ±0.16 | 1.01 ±0.18 | 0.1797 ±0.0481 |