# OpenReview forum: "Neural Clamping: Joint Input Perturbation and Temperature Scaling for Neural Network Calibration"
_TMLR — Accepted by TMLR_

### Review · Reviewer_GNEb · 2024-05-17

**Summary Of Contributions:**

The paper introduces Neural Clamping, a novel post-processing calibration method for deep neural networks. This method enhances model calibration by jointly applying a learnable universal input perturbation and an output temperature scaling parameter to a pre-trained classifier. Theoretical proofs are provided to demonstrate the superiority of Neural Clamping over traditional temperature scaling. Empirical evaluations on datasets like BloodMNIST, CIFAR-100, and ImageNet reveal that Neural Clamping significantly improves calibration performance across various deep neural network architectures, achieving up to 34% better expected calibration error compared to state-of-the-art methods.

**Audience:**

Yes

**Broader Impact Concerns:**

The paper introduce a new framework of calibration, which is motivative.

**Claims And Evidence:**

Yes

**Requested Changes:**

1. add more recent related works such as [1] [2]
2. Compare to recent proposed post-processing works
3. Answer the question in above section



[1] Dual Focal Loss for Calibration
[2] Proximity-Informed Calibration for Deep Neural Networks

**Strengths And Weaknesses:**

**Strengths**

The introduction of noise perturbation is a good try. Very surprise  to see it works together with temperature scaling on calibration task.

---

**Weaknesses**

1. The motivation is not clear. Why adding a noise together with the temperature scaling will benefit calibration? Better to adding more details or toy examples to illustrate the motivation in the introduction part.
2. What is the calibration set mentioned in introduction? Is it a validation set? not clear in introduction
3. Does it have to be focal loss to train the noise and temperature parameter? not clear in introduction
4. In related works, did not mention the recent works in in-processing calibration methods such as [1]
5. Lack of analysis and comparison with recent post-processing methods such as [2]
6. Lack of constrain of perturbation, should be mentioned earlier than eq 7. Is it like the adversarial samples?
7. It is not clear how you choose gamma in 3.3. Do you split another set for choosing gamma? Or using the same "calibration set"
8. The introduction of noise perturbation is a good try. Very surprise  to see it works together with temperature scaling on calibration task. Can the author give more analysis on the ablation study why noise along does not work well.






[1] Dual Focal Loss for Calibration
[2] Proximity-Informed Calibration for Deep Neural Networks

---

> ### Author Response · Authors · 2024-06-23
> **Response to Reviewer GNEb**
>
> Thank Reviewer GNEb for the insightful comments and suggestions. We are delighted to receive the positive comment that “Very surprise to see it works together with temperature scaling on calibration task.” We appreciate your review and would like to provide some clarification.
>
> -----------------
>
> ### [Q1] Why input perturbation can benefit calibration? (weakness 1 & 8)
> The combination of input and output calibration methods is crucial to maximizing overall calibration performance. As shown in Theorem 3.2 in Section 3.4, input perturbation can maximize the output entropy of a neural network. Furthermore, input perturbation directly influences the representations of the logit values, enhancing the distinction between the class logits. This enhanced adjustment in the logit space can then be further amplified by the output calibration via temperature scaling. It is noteworthy that the joint input-output calibration approach has been shown to reduce ECE and AECE by an additional 25% compared to output calibration alone in Table 4. By synergistically leveraging both techniques, the model's ability to differentiate between classes is significantly strengthened, leading to notable improvements. The complementary effect between the two calibration approaches is the key to achieving the best calibration results.
>
> ### [Q2] What is the calibration set? Is it a validation set?
> Yes, as stated in the experimental setup in Section 4.1, the calibration set is the same as the validation set used during model training. We follow the terminology commonly used in other calibration-focused papers to refer to this dataset as the "calibration set". We've added some information about this confusion in the introduction, hoping to make it clearer.
>
> ### [Q3] Does it have to be focal loss to train the noise and temperature parameters?
> No, it does not have to be focal loss to train the noise and temperature parameters. In our experiments, we have compared the results using both cross-entropy loss (a special case of focal loss) and focal loss as the training loss, and our method outperforms other baselines in both cases. However, we found that the combination of our method with focal loss can achieve even better calibration performance.
>
> ### [Q4] Did not mention the recent works in in-processing calibration methods such as [1].
> We have added references to recent in-processing calibration methods, such as [1], in the revised version.
>
> ### [Q5] Lack of analysis and comparison with recent post-processing methods such as [2].
> Yes, this is another post-processing method to calibrate models. However, this method is limited to calibration on the top-1 class prediction but cannot be extended to all-class predictions.
> Nevertheless, we have briefly mentioned the method [2] in the revised version and highlight its limitations, in order to emphasize the advantages of our method in all-class calibration.
>
> ### [Q6] Lack of constraint of perturbation, should be mentioned earlier than eq 7. Is it like the adversarial samples?
> No, we don't need to limit the perturbation length. Our goal is calibration, not generating inputs to fool the model. By not restricting perturbations, the input perturbation can freely adjust the model's output probabilities, enabling more effective alignment between confidence and true accuracy.
>
> ### [Q7] It is not clear how you choose gamma in 3.3. Do you split another set for choosing gamma? Or using the same "calibration set"
> As mentioned in the last sentence of Section 3.3, we select the value of gamma that minimizes the Expected Calibration Error (ECE) on the calibration dataset.
> Specifically, we do not split the dataset further to choose gamma. Instead, we use the same calibration set to evaluate the ECE for different candidate values of gamma. We then pick the gamma that gives the lowest ECE on this calibration set.

---

### Review · Reviewer_7Ruf · 2024-05-25

**Summary Of Contributions:**

The paper presents neural clamping, a generalization of temperature scaling scaling which adds a fixed, trainable, additive perturbation to the input of the model at hand, with both the perturbation and temperate fitted with focal loss instead of basic CE.

The method is evaluated on various datasets and model architectures and appears to improve ECE, AECE and SCE, sometimes also slightly improving accuracy. A theoretical analysis in terms of entropy maximisation and a data-derived init for the perturbation are also presented

**Audience:**

Yes

**Broader Impact Concerns:**

No need for broader impact assessment.

**Claims And Evidence:**

No

**Requested Changes:**

# Critical

1. run an ablation where you fit the temperature only with the focal loss - currently either temperature alone is fitted with CE or temperature+input are fitted with Focal loss,  but then only one is used
2. pick at least one dataset (I'd propose cifar100) and train each architecture studied on it (multiple seeds for training), then perform the above ablation and your other tests and then perform a statistical significance test (e.g. welch test) or another rigorous assessment across independently trained models (I suggest also using multiple independent calibration datasets on each model)


without these, I can't call them method rigorously supported
## Nice to have
1. above on all datasets => but I understand computational expense might be limiting
2. if you can keep snapshots across training, you can try the calibration on various stages of undertraining to assess the importance of model capacity and fittedness

**Strengths And Weaknesses:**

\+ interesting idea to modify both input and output, likewise to use the focal loss to tune
\+ method appears to perform well across benchmarks
\- experiments appear to be done on single pretrained models? would require multiple seeds *on models* to be meaningful
\- having the same architectures across datasets would also make analysis better
\- also, statistical significance test should be done since the methods are sometimes quite close
\- I have the suspicion that the benefit mainly derives from the focal loss and not from the input perturbation => should be done in an ablation
\- slight test leakage required for the accuracy to improve (effectively, we are learning a slighlty adjusted initial centering operation)
\- nitpick: english is weird in some place s("Uncalibration" instead of "Uncalibrated" in tables,"methods predominently shed" "confidence made by model prediction")

---

> ### Author Response · Authors · 2024-06-23
> **Response to Reviewer 7Ruf**
>
> Thanks Reviewer 7Ruf for these insightful comments and suggestions. We are delighted to receive the positive comment that “interesting idea” and  “method appears to perform well across benchmarks.” We have responded to your questions and made some modification in the revised version:
>
> --------------
>
> ### [Q1] Statistical significance tests should be done.
> We have added the p-value and t-statistic of our method and second-best baseline method in the appendix in the revised version.
> We check all experiments in our main paper, our proposed method (Neural Clamping) shows extremely statistically significant differences to the corresponding second-best baseline method (such as Dir-ODIR, TS(GS), TS, etc.).
> The p-values are extremely low, typically less than 0.001 significance level, and some even less than 0.00001. This means the observed differences are highly unlikely to have occurred by random chance.
> The absolute values of the t-statistics are very high, often above 10, and some even exceeding 80. This indicates the performance gap between the two methods is very large.
> These statistical metrics provide very strong evidence that your proposed NC (FL) method significantly outperforms the baseline methods across various datasets and network architectures.
>
>
> ResNet-50 @ BloodMNIST: NC(FL) vs. Dir-ODIR:
> - t-statistic: 22.36
> - p-value: 0.000000044
>
> ResNet-110 @ CIFIAR-100: NC (FL) vs. TS(GS)
> - t-statistic: 17.14
> - p-value: 0.000067
>
> Wide-ResNet-40-10 @ CIFAR-100: NC (FL) vs. TS (GS)
> - t-statistic: 23.88
> - p-value: 0.000018
>
> ResNet-101 @ ImageNet-1K: NC (FL) vs. TS
> - t-statistic: 83.41
> - p-value: 0.00000012
>
> ViT-S/16 @ ImageNet-1K: NC (FL) vs. TS (GS)
> - t-statistic: 11.18
> - p-value: 0.00036
>
> MLP-Mixer B/16 @ ImageNet-1K: NC (FL) vs. TS (GS)
> - t-statistic: 5.68
> - p-value: 0.0047
>
>
> ### [Q2] slight test leakage required for the accuracy to improve.
> We use the validation dataset for calibration, while the test dataset is only used for final evaluation. This is  a common approach for calibration research, such as Guo et al., Kull et al.
>
> ### [Q3] run an ablation where you fit the temperature scaling only with the focal loss.
> We add temperature scaling with focal loss in ablation study in the revised version (Table 4. in main paper, below Table S1 for briefly review). We use the same method to select gamma for temperature scaling (FL). Firstly, the "Temperature Scaling (FL)" approach, where temperature scaling is applied with focal loss, actually performs worse than both the original "Temperature Scaling" method and "Output Calibration w/ T*". Across the various calibration metrics, including Accuracy, ECE, AECE, and SCE, the "Temperature Scaling (FL)" ablation shows poorer performance compared to the other methods.
> This finding suggests that the improvements achieved by our method Neural Clamping (joint input-output calibration) is not simply due to the use of temperature scaling with focal loss. There appear to be additional benefits gained from the joint calibration framework that go beyond just temperature scaling with focal loss.
>
> **Table S1. ablation study with ResNet-110 on CIFAR-100 (Table 4. in main paper).**
> | Method                        | Accuracy (%) | Entropy | ECE   | AECE  | SCE    |
> |-------------------------------|--------------|---------|-------|-------|--------|
> | Uncalibrated                  | 74.15        | 0.4742  | 10.74 | 10.71 | 0.2763 |
> | Temperature Scaling           | 74.15        | 0.8991  | 1.71  | 1.63  | 0.1711 |
> | Temperature Scaling (FL)      | 74.15        | 1.0542  | 2.01  | 2.02  | 0.1812 |
> | Input Calibration w/ $\delta$ | 74.16        | 0.4775  | 10.62 | 10.61 | 0.2776 |
> | Output Calibration w/ $T$     | 74.15        | 0.9648  | 1.18  | 1.35  | 0.1731 |
> | Neural Clamping               | 74.16        | 0.9707  | 0.89  | 1.01  | 0.1754 |

---

> ### Author Response · Authors · 2024-06-23
> **Response to Reviewer 7Ruf (cont.)**
>
> ### [Q4] Evaluate architecture with multiple training seeds, then perform the ablation (temperature + focal loss) and other tests, and conduct a statistical significance test (e.g., Welch's t-test).
> We trained ResNet-110 on CIFAR-100 with 5 random initialization seeds to account for potential variability in the training process (Table 12 in appendix, below Table S2 for briefly review). For each of the 5 trained models, we performed the post-training calibration using a 5-fold cross-validation approach. This means that for each of the 5 trained models, we obtained 5 calibration datasets, resulting in a total of 25 calibration runs (5 models × 5 folds).
> Then, we conducted a Welch's t-test to compare our proposed method (Neural Clamping) and the second-best baseline method (TS by Grid Search). The results show:
> - p-value = 0.00000000000004
> - t-statistic = 10.67
>
> The extremely small p-value (well below the 0.05 significance level) indicates that there is a highly statistically significant difference between the Neural Clamping method and the TS by Grid Search method. The absolute value of the t-statistic, 10.67, is far greater than the critical value of 2, further confirming the highly significant difference (reduces 35% ECE)  between the two methods.
>
> In summary, experimental results demonstrate that the Neural Clamping method has a significantly superior calibration performance compared to other baseline methods.
>
> **Table S2. Comparison with various calibration methods on CIFAR-100 with ResNet-110 with different
> training seeds (Table 12 in appendix)**
> | Method                   | Accuracy (%) | Entropy | ECE   | AECE  | SCE    |
> |--------------------------|--------------|---------|-------|-------|--------|
> | Uncalibrated             | 73.95        | 0.5041  | 10.14 | 10.12 | 0.2722 |
> | Temperature Scaling      | 73.95        | 0.9146  | 1.75  | 1.56  | 0.1776 |
> | Temperature Scaling (FL) | 73.95        | 1.0777  | 2.35  | 2.35  | 0.1889 |
> | TS by Grid Search        | 73.95        | 0.9476  | 1.49  | 1.27  | 0.1480 |
> | Vector Scaling           | 73.45        | 0.8738  | 2.27  | 2.11  | 0.1799 |
> | Matrix Scaling           | 63.95        | 0.1648  | 31.71 | 31.71 | 0.6813 |
> | MS-ODIR                  | 73.58        | 0.9187  | 1.83  | 1.51  | 0.1801 |
> | Dir-ODIR                 | 73.78        | 0.9504  | 1.58  | 1.32  | 0.1762 |
> | Neural Clamping (CE)     | 74.16        | 0.9514  | 1.06  | 1.14  | 0.1634 |
> | Neural Clamping (FL)     | 74.17        | 0.9921  | 0.96  | 1.01  | 0.1797 |
>
>
> ### [Q5] if you can keep snapshots across training, you can try the calibration on various stages of undertraining to assess the importance of model capacity and fittedness.
> Unfortunately, since the large models we are working with are pre-trained and obtained from GitHub, we do not have access to these training checkpoints. This limits our ability to explore the relationship between model capacity, fittedness, and calibration performance at different stages of the training process. We hope our results on ResNet-110 on CIFAR-100 in the previous response address the reviewer’s concern.

---

### Review · Reviewer_pRL2 · 2024-06-09

**Summary Of Contributions:**

This paper presents a new calibration method that combines input perturbation with temperature scaling. The prior method only considers using temperature scaling to modify output logits, while this work proposes using a learnable universal input perturbation to improve the calibration without changing the model parameters. It has theoretical result about optimal perturbation value and the improved entropy (calibration-related metric). It has empirical results in improving calibration metrics such as ECE in ResNets, Transformers and CIFAR, ImageNet datasets.

**Audience:**

Yes

**Broader Impact Concerns:**

The border impact has been discussed in the impact section, and no concerns regarding that.

**Claims And Evidence:**

Yes

**Requested Changes:**

I would like to ask for some clarification.

1. If I understand correctly, the $\widetilde{\boldsymbol{\delta}}$ is derived from Theorem 3.2 and its closed form is $\operatorname{sign}(\boldsymbol{g}) \odot \boldsymbol{\eta}$, the $\widetilde{\boldsymbol{\delta}}$ is used as "data-driven initialization", $\boldsymbol{\delta}$ is then trained. Could the authors provide additional details, such as an algorithm diagram or text description, to explain how the "data-driven initialization" is implemented?
2. What is the rationale for using the closed-form optimal value $\widetilde{\boldsymbol{\delta}}$ that maximizes entropy as an initialization? If the theoretical result of Theorem 3.2 is correct, then one might suppose that the derived $\widetilde{\boldsymbol{\delta}}$ can be directly used at test time (since it maximizes entropy). How is that compared to $\boldsymbol{\delta}$ after training, or randomly selected value in evaluations of Figure 3?
3. Some of the theoretical results are confusing. Was Lemma 3.1 and its proof a replicate of Appendix S.2 of [Guo et al](https://arxiv.org/pdf/1706.04599) with $\mathbf{x}$ changed to $\mathbf{x} + \boldsymbol{\delta}$. I am confused about what are the new insights/differences between the presented one and the one in prior work.

**Strengths And Weaknesses:**

### Strengths
1. The empirical evaluation is comprehensive, it studies three datasets and four architectures, and compares with about 7 baselines. The effectiveness of the proposed method as a good calibration method seems to be pronounced.
2. The paper is well-motivated, it focuses on post-processing, which improves calibration without changing the model parameters, similar to prompt engineering in LLM.
3. The paper is overall well-written, with preliminary definitions carefully stated.

### Weakness
1. The novelty of the proposed method is a concern. The improvement in calibration through input perturbation has been studied by [Qin et al](https://arxiv.org/abs/2006.16375) and temperature scaling is a traditional calibration approach. The technical novelty of the paper lies in making both components trainable at the post-training stage without changing the model parameters.
2. The results in Table 4 suggest that input calibration may not be as necessary as output calibration, which raises questions about the authors' claim that "This ablation study corroborates the necessity and advantage of joint input-output calibration."
The ablation study of Section 4.3, Table 4 shows that the ECE is reduced by 9.54 with output calibration (temperature scaling), but is reduced only by 0.1 - 0.3 with input calibration (perturbation).

---

> ### Author Response · Authors · 2024-06-23
> **Response to Reviewer pRL2**
>
> Thank Reviewer pRL2 for the valuable feedback on the novelty of our proposed method and some questions about our data-driven delta and theoretical result. We are delighted to receive the positive comment that “empirical evaluation is comprehensive”, “well-motivated”, and “well-written.” We appreciate your review and would like to provide some clarification.
>
> -------------
>
> ### [W1] The novelty of the proposed method is a concern.
> The key novelty of our method lies in the joint optimization of input perturbation and temperature scaling at the post-training stage, without modifying the model parameters.
> We acknowledge that the relationship between adversarial robustness and calibration has been studied by Qin et al. However, their approach of using label smoothing at different degrees based on adversarial robustness is an in-processing method, which is fundamentally different from our post-processing approach. Importantly, Qin et al.'s work did not explore the use of input perturbation to improve calibration, which is a central component of our proposed method.
> We have mentioned Qin et al.’s work in the revised version.
>
> ### [W2] Concern about the necessity of joint input-output calibration.
> The combination of input and output calibration methods is crucial to maximizing overall calibration performance. Input calibration via input perturbation directly influences representations of the logit values, enhancing the distinction between class logits. This enhanced adjustment in the logit space can then be further amplified by the output calibration via temperature scaling. It is noteworthy that the joint input-output calibration approach has been shown to reduce ECE and AECE by an additional 25% compared to output calibration alone in Table 4. By synergistically leveraging both techniques, the model's ability to differentiate between classes is significantly strengthened, leading to notable improvements. The complementary effect between the two calibration approaches is the key to achieving the best calibration results.
>
> ### [Q1] Could the authors provide more details on how the "data-driven initialization" using the closed-form $\widetilde{\boldsymbol{\delta}}$ is implemented?
> According to Theorem 3.2 in the paper, data-driven initialization starts from $ \widetilde{\delta} = \textsf{sign}(g) \odot \eta$. In contrast, random initialization means filled $\delta$ with random numbers from a normal distribution with mean 0 and small variance.
>
> For the g term, we use input data and labels from the validation dataset to compute the input gradient. Specifically: (1) The input data from the validation set is wrapped in a PyTorch Variable object, with requires_grad=True. This enables PyTorch to automatically compute the gradients with respect to this input variable during the subsequent backpropagation step. (2) A forward pass is then performed through the model, taking the input data as input and producing the output logits. (3) The torch.autograd.grad function is used to calculate the gradient of the output logits with respect to the input data. The grad_outputs parameter is set appropriately to indicate the desired gradient outputs. (4) All the computed input gradients are then collected into a single tensor, denoted as sign(g), which represents the sign of the input gradients.
>
> For the η term, one can simply calculate the upper $\mu$ and lower bounds $\ell$ for each pixel from the validation dataset. Then η values can be easily calculated from $\mu$ and $\ell$ according to input gradient g, in order to maximize the scalar product δ⊤g.
>
>
> ### [Q2] What is the rationale for using the theoretically optimal $\widetilde{\boldsymbol{\delta}}$ as an initialization, and how does it compare to the trained $\boldsymbol{\delta}$ or randomly selected values?
>
> The rationale for using the theoretically optimal $\widetilde{\boldsymbol{\delta}}$ as an initialization is that our theoretical analysis proves this value is the optimal solution to the  first-order approximation of the objective function.
>
> As shown in ablation study, compared to training with a randomly initialized $\boldsymbol{\delta}$, using the first-order approximate $\widetilde{\boldsymbol{\delta}}$ as the starting point can lead to more stable and reliable outcomes. This suggests that this first-order approximation is already sufficiently close to the optimal solution.
>
> In addition, based on the study reported in the appendix (Computationally Efficient Neural Clamping), even this approximate theoretically optimal solution $\widetilde{\boldsymbol{\delta}}$ already performs very close to the final trained optimal $\boldsymbol{\delta}$. This suggests that our theoretical analysis, though only a first-order approximation, has already captured the key essence of the problem.

---

> ### Author Response · Authors · 2024-06-23
> **Response to Reviewr pRL2 (cont.)**
>
> ### [Q3] Is Lemma 3.1 and its proof simply a replication of Guo et al.'s Appendix S.2 with $\mathbf{x}$ changed to $\mathbf{x} + \boldsymbol{\delta}$, and what are the new insights/differences compared to the prior work?
>
> Our Lemma 3.1 and its proof are a more general extension of the results presented by Guo et al. in their Appendix S.2. Guo et al. considered the case of output scaling, while we have generalized their result to the case of input perturbation and output scaling.
>
> This extension makes our result more broadly applicable, covering not only the special case of temperature scaling, but also a wider range of $\boldsymbol{\delta}$ input perturbation techniques. Our proof follows a similar line of reasoning as Guo et al., but is able to derive a more generally applicable conclusion.

---

### Decision · Action_Editor_2RFS · 2024-07-15

**Recommendation:** Accept with minor revision

**Comment:**

While there were some mixed impressions concerning the novelty of the method and the evaluation methods, these concerns were clarified during the discussion phase and changes were made in the manuscript. After the rebuttal, all reviewers recommend the acceptance of the paper at TMLR upon the following minor revisions:

- Reviewer *7Ruf* noted that "*the t-statistic and p value are possibly slightly misleading on the current stage, as they assume that the 25 samples are IID which one could argue is not likely (as they are derived from the same models). I think all that is required to fix this is to tweak the language in the camera ready and/or make an argument for why the authors chose this particular setup*"

- Reviewer *pRL2* noted that: "*based on the rebuttal, it is encouraging to know that the theoretically derived input perturbation can achieve performance similar to that of training results. I suggest highlighting this in the paper. Lastly, I still don't think that Lemma 3.1 and its proof made a significant contribution.*"

Therefore, I encourage the authors to implement these changes and I am happy to recommend the paper for publication at TMLR.

**Audience:**

I believe this work is of interest to the TMLR audience interested in uncertainty quantification methods for neural networks.

**Claims And Evidence:**

This work proposes a new calibration method combining input perturbations and temperature scaling, and provides empirical evidence that their method improve the ECE with respect to the literature benchmarks, accross a variety of data sets and architectures. The paper also provides a theoretical result on when their method outperforms temperature scaling (Theorem 3.2).

---

> ### Author Response · Authors · 2024-07-19
> **Response to Reviewers' Comments and Revised Manuscript Submission**
>
> Dear Editor,
>
> We sincerely appreciate the reviewers' thorough evaluation and valuable feedback on our manuscript. We are grateful for their constructive comments and recommendations, which have significantly contributed to improving our work.
>
> We have made the following revisions based on the reviewers' suggestions:
> - We added the code link and demo link in the abstract.
> - Following Reviewer pRL2's suggestion, we highlighted in Section 4.3 that theoretically derived input perturbation can achieve performance similar to that of training results.
> - Following Reviewer 7Ruf's suggestion, we explained the reasons for our experimental setup in Appendix G.
>
> We are encouraged by the positive feedback and are grateful for the opportunity to revise our manuscript. We look forward to seeing our work published in TMLR.
>
> #2601 Authors